# Calicivirus assembly and stability are mediated by the N-terminal domain of the capsid protein with the involvement of the viral genome

Guy Novoa[1][☉], Carlos P. Mata[1][☉][¤a], Johann Mertens[2], María Zamora-Ceballos[3],
Juan M. Martínez-Romero[1], Juan Fontana[4][¤b], José L. Carrascosa[1,5], Juan Bárcena[3]*,
José R. Castón[1,5]*

1 Department of Structure of Macromolecules, Centro Nacional de Biotecnología (CNB-CSIC), Campus de Cantoblanco, Madrid, Spain, 2 Instituto Madrileño de Estudios Avanzados en Nanociencia (IMDEA Nanociencia), Campus Cantoblanco, Madrid, Spain, 3 Centro de Investigación en Sanidad Animal (CISA-INIA/CSIC), Madrid, Spain, 4 Faculty of Biological Sciences and Astbury Centre for Structural and Molecular Biology, School of Molecular and Cellular Biology, University of Leeds, Leeds, United Kingdom, 5 Nanobiotechnology Associated Unit CNB-CSIC-IMDEA, Campus Cantoblanco, Madrid, Spain

☉ These authors contributed equally to this work.
¤a Current address: PKF Attest innCome, Madrid, Spain
¤b Current address: Instituto Biofisika, CSIC-UPV/EHU, Barrio Sarriena s/n, Leioa, Bizkaia, Spain
* barcena@inia.csic.es (JB); jrcaston@cnb.csic.es (JRC)

## Abstract

Caliciviruses are important human and animal pathogens that cause varying clinical signs including gastroenteritis, respiratory illness, and hepatitis. Despite the availability of numerous calicivirus structures, relatively little is known about the mechanisms of capsid assembly and stability, or about genome packaging. Here we present the atomic structure of the RHDV virion and several related non-infectious virus-like particles, determined using cryo-EM at 2.5-3.3 Å resolution. The inherent molecular switch, responsible for the conformational flexibility of the capsid protein VP1, is located in its N-terminal arm (NTA). The NTA establishes an extensive network of interactions on the inner capsid surface that stabilizes the hexamers and pentamers. For this structural polymorphism, we show that the NTA must interact with the RNA viral genome, that is, the genomic RNA acts with the NTA as a molecular co-switch. The NTA-RNA interaction leads to specific conformational states that result in two types of VP1 dimers (the basic building blocks) necessary for T = 3 capsid assembly. In addition, we used atomic force microscopy (AFM) to assess whether differences in genomic RNA content influence viral properties such as capsid stiffness in physiological conditions. These analyses highlight the mechanical role of packed RNA genome in RHDV virions, as the virion capsid pentamers are strengthened by interactions of the NTA star-like structure promoted by the viral genome. These results indicate that the interactions between the NTA and the viral genome guide the conformational states of VP1 dimers, directing capsid assembly and modulating its mechanical properties. Through interference with intermediate assemblies, the NTA network promoted by the genome could be an attractive target in future antiviral strategies.

the Creative Commons Attribution License, which permits unrestricted use, distribution, and reproduction in any medium, provided the original author and source are credited.

**Data availability statement:** The atomic coordinates and cryo-EM density maps were deposited in the Protein Data Bank and EM Data Bank with codes 9I9D and EMD-52757 for the full RHDV virion, 9B8I90 and EMD-52747 for the empty RHDV capsid, 9I3H and EMD-52593 for the N15 capsid, 9I8R and EMD-52733 for the VP1 capsid, 9I3E and EMD-52588 for the Δ29N capsid, and 9I8J and EMD-52727 for the P domain of N15 A/B subunits. All other relevant data are available in the main text or the supplementary materials.

**Funding:** This work was supported by grants from the Spanish Ministry of Science and Innovation (PID2020-113287RB-I00 and PID2023-146143NB-I00) to J.R.C., PID2022-140925OB-I00 to J.B., and PID2023-149259NB-I00 to J.F. M.Z-C held a PhD fellowship from the Spanish Ministry of Science and Innovation (BES-2017-081188). The CNB-CSIC was further supported by an institutional grant from the Severo Ochoa Program for Centers of Excellence in R&D (CEX2023-001386-S/AEI/10.13039/501100011033). The funders had no role in study design, data collection and analysis, decision to publish, or preparation of the manuscript.

**Competing interests:** The authors have declared that no competing interests exist.

## Author summary

Rabbit hemorrhagic disease virus (RHDV) is a highly contagious, often fatal calicivirus that poses a significant threat to rabbit populations worldwide. Although considerable progress has been made in elucidating the structural organization of RHDV capsids at near-atomic resolution, information regarding the nucleic acid organization within these virions and its interactions with the viral capsid remains limited. Understanding these interactions is essential for identifying mechanisms of structural polymorphism of the capsid protein, as well as virion assembly and stability. We report the cryo-electron microscopy structure, at near-atomic resolution, of RHDV virions and distinct virus-like particles, together with characterization of their biophysical properties using atomic force microscopy. The N-terminal region of the capsid protein, essential for its structural polymorphism, requires its interaction with genomic RNA. The interaction between N-terminal arms and the viral genome governs the conformational states of capsid protein dimers, thereby orchestrating capsid assembly and influencing its biophysical properties. By interfering with intermediate assemblies, the NTA network promoted by the genome could constitute an attractive target in antiviral strategies for several major human calicivirus pathogens that cause acute gastroenteritis and respiratory illness, including norovirus and sapovirus.

## Introduction

The genus *Lagovirus* within the *Caliciviridae* family [1] comprises viruses that infect leporids (rabbits and hares). It includes highly pathogenic viruses such as rabbit hemorrhagic disease virus (RHDV), which causes fatal hepatitis in European rabbits (*Oryctolagus cuniculus*), with case fatality rates >90% among infected animals [2,3]. RHDV, which belongs to genotype GI.1 according to the current lagovirus nomenclature [4], emerged in domestic rabbits in China in 1984, whence it rapidly spread to Asia, Europe, Central America, and Africa, causing vast economic damage to rabbit farming as well as negative environmental impacts [5]. RHDV was subsequently imported to Australia in 1996 as a biocontrol agent of feral rabbit populations [6]. In 2010, a new pathogenic lagovirus with distinctive biological and antigenic features, RHDV2 (genotype GI.2), appeared in France [7] and spread worldwide, affecting not only wild and domestic European rabbit populations, but also several hare species, including endangered ones [8,9]. Both pathogenic lagoviruses, RHDV and RHDV2, have caused significant ecological and economic damage, representing a serious threat to the European rabbit and the related industry [10,11].

Lagoviruses are nonenveloped icosahedral viruses with a positive-sense, single-stranded RNA (+ssRNA) genome of ~7.4 kb and a 3′ poly-A tail, organized in two overlapping open reading frames (ORF) flanked by untranslated regions (UTR) at both the 5′ and 3′ ends. ORF1 encodes several non-structural proteins (helicase, VPg, protease, and RdRp among others), as well as the major structural protein VP1

(formerly VP60). ORF2 encodes a minor structural protein, VP2, which is involved in release of the viral genome into the infected host cell [12]. VP1 and VP2 are also expressed from a subgenomic RNA~2.4 kb [13].

The family *Caliciviridae*, which includes several major human pathogens, is currently divided into eleven genera [1]. Members of *Lagovirus*, *Norovirus*, *Nebovirus*, *Recovirus*, *Sapovirus*, *Valovirus*, and *Vesivirus* infect mammals; members of *Bavovirus* and *Nacovirus* infect birds, and members of *Minovirus* and *Salovirus* infect fish. A number of X-ray crystallography and cryo-electron microscopy (cryo-EM) atomic structures of caliciviruses have been determined using virus-like particles (VLP) or virions of *Lagovirus* [14,15], *Norovirus* [16–20], *Sapovirus* [21] and *Vesivirus* [12,22,23].

For RHDV, the structure of a pseudo-atomic model of the RHDV GI.1 capsid was calculated by combining the crystal structures of VP1 domains with low resolution cryo-EM structures [14], as well as the cryo-EM structure of RHDV GI.2 virions [15]. Calicivirus capsids (~40 nm diameter) have a common capsid shell organization, which is assembled by 180 copies of VP1, organized as 90 dimers that form arch-like capsomers arranged with T = 3 symmetry to form 12 pentamers and 20 hexamers [24,25]. The icosahedral asymmetric unit is formed by three quasi-equivalent conformations of VP1 termed A, B, and C, that is, identical protein subunits participate in three different molecular interactions. In icosahedral viruses, these differences are quite subtle and are controlled by flexible regions in the protein (loops, N ends, and C ends), double- or single-stranded RNA, metal ions, or some combination of these [26,27]. These factors that direct the capsid assembly are referred to as molecular switches.

A, B and C conformers of VP1 are organized in 60 A/B dimers and 30 C/C dimers [16]. Each VP1 monomer has three domains [14,15], an internally located N-terminal arm (NTA), a shell domain (S) composed of an eight-stranded β-sandwich, which forms a continuous scaffold protecting the viral RNA, and a flexible protruding domain (P) at the capsid surface, involved in virus–host receptor interactions and antigenic diversity [28–30]. The P domain is subdivided into P1 and P2 subdomains. The P2 subdomain, located at the outermost surface of the viral capsid, contains seven loops of various lengths that exhibit the highest degree of sequence variation among RHDV strains [14].

RHDV, unlike some other caliciviruses, is unable to replicate in cell culture systems. Many studies on these viruses are performed using recombinant VLP, which are morphologically and antigenically similar to authentic RHDV virions [31]. RHDV VLP were shown to induce full protection of rabbits against a lethal challenge with RHDV [32], and are used as diagnostic reagents [33] as well as a vaccine platform for multimeric antigen display [34,35].

Although extensive structural data are available for RHDV capsids, information regarding the spatial organization of the ssRNA within these virions and its interactions with the capsid is lacking. Here we report the three-dimensional (3D) cryo-EM structures of the RHDV GI.1 virion grown in infected rabbits, as well as distinct VP1-derived VLPs, both at 3.3-2.5 Å resolution. The molecular switch for VP1 conformational flexibility is located at the NTA N-terminal end; it has an ordered structure within the hexameric assemblies, but is disordered within the pentameric assemblies. The switch acts as a wedge that prevents pivoting of the protein surfaces in the hexamers. The NTA nevertheless requires interaction with genomic RNA, which also acts as the molecular co-switch. This NTA-RNA interaction leads to specific conformational states of VP1 dimers in the T = 3 capsid, to direct the capsid assembly and modulate the capsid mechanical properties.

## Results

### Biochemical and structural analysis of the RHDV virion and related VLPs

RHDV GI.1 virions (AST89 strain), obtained from the livers of infected rabbits, were purified by rate zonal centrifugation in a sucrose density gradient. We also formed VLPs from recombinantly expressed VP1-related proteins with modifications at the N-terminal end, including (i) the wild-type (wt) version of 579 residues [24], (ii) the N15 chimeric protein that bears a foot-and-mouth disease virus (FMDV)-derived T-cell epitope (15 residues) [36] joined at the VP1 N terminus, which assembled into chimeric VLPs displaying a foreign epitope to be used as a vaccine platform [37], and (iii) the Δ29N deletion mutant that lacks the first 29 N-terminal residues [24] (Fig 1A). These purified virions and VP1-related assemblies resulted in single bands in SDS-PAGE analysis (Fig 1B).

PLOS Pathogens

Cryo-EM of purified virions showed a mixture of full and empty ~41-nm-diameter particles (Fig 1C). Cryo-EM of VLP-enriched fractions showed that N15 and wt VP1 capsids are similar to virions in size and morphology (Fig 1D and 1E), whereas Δ29N VLPs measured ~28 or ~41 nm in diameter (Fig 1F). We calculated icosahedrally averaged maps at 2.5-3.3 Å resolution for these capsids, as estimated by the criterion of a 0.143 Fourier shell correlation (FSC) coefficient (Fig 1G and S1 Table). The molecular architecture of the virion, N15, and wt VP1 capsids is essentially as described previously [14,25], a T = 3 capsid with 90 dimeric protrusions that surround 12 pentameric and 20 hexameric depressions at the 5- and 3-fold axes, respectively (Fig 1H–J, top). The asymmetric unit is formed by three quasi-equivalent conformations of identical subunits termed A, B, and C, which are defined by the occupancy of structurally distinct environments. Subunits A, B, and C cluster into two dimer classes, A/B and C/C, as described for other calicivirus capsids and plant viruses such as tomato bushy stunt virus [38] and rice yellow mottle virus [39]. The small Δ29N capsids showed a T = 1 lattice with 30 protruding dimers located at the icosahedral twofold axis. Dimeric protrusions were conformationally identical, but they are formed by a structural subunit different from those forming the T = 3 capsid (labeled A').

The resolution of different regions of the cryo-EM maps was determined using the MonoRes program [40]. Whereas resolution of the continuous shell of the T = 1 and T = 3 capsids was high and homogeneous, the protrusions in their outermost zones were the regions with the lowest resolution, particularly in the A/B and A'/A' dimers, indicating their high flexibility (Figs 1H–K, bottom, and S1).

## Cryo-EM near-atomic model of the N15 chimeric capsid P domain

Analysis with 3DFlex [41] of T = 1 Δ29N capsids and by 3D classification of Δ29N extracted dimers allowed us to determine that the protrusion swinging angle ranges between ~10º and ~27º (S1 Video and S2A Fig). In the T = 1 capsid, the direction of the swing is perpendicular to the orientation of the hinge regions within the dimer (S2B Fig). These results suggest a unique, allowed movement of the spikes.

Due to the high flexibility of protrusion domain, Δ29N protein was modeled only for the shell domain. 3DFlex analysis of T = 3 capsids (full and empty virions) similarly rendered limited motions of ~5º or less for protrusions (S2 and S3 Videos, respectively). N15 capsids were appropriate to solve the native structure of the P domain (below).

Comparison of sharpened maps for C/C (Fig 2A, right) and A/B (Fig 2B, right) protruding domains, calculated after icosahedral averaging of N15 capsids, showed that C/C dimers were at an appropriate resolution to build the polypeptide chain (between 2.5-3.5 Å at the tip region based on the local resolution assessment, Fig 1I, lower row); A/B dimers, particularly the uppermost region, were too noisy to trace the backbone with numerous disconnected densities (between 3–5 Å resolution, Fig 1I, lower row). A/B dimers of the N15 capsid were selected and extracted from the particle images as independent subparticles, then subjected to 3D classification without angular assignment. As a result, > 164K particle images were selected and combined to yield a 2.9-Å resolution map (Fig 2C). This procedure only worked well for N15 particle images (not for other T = 3 particle images), probably because the N15 dataset was acquired with a Titan Krios electron microscope equipped with a K2 Summit direct electron detector in counting mode (see S1 Table). The resulting A/B protrusion after local reconstruction was locally sharpened with LocalDeblur [42], and the atomic model was built *de novo*, given that most amino acid side chains were resolved in the 2.9-Å resolution cryo-EM density (Fig 2D and 2E). Our cryo-EM structure of P domain dimer and the X-ray structure of recombinant P domain dimer [14] show similar structures, in which the mean Cα - Cα distances (rmsd) were ~1 Å for equivalent Cα after superimposition of 295 P residues (of 333).

## Cryo-EM near-atomic model of the RHDV virion

Of the total 579 residues, the cryo-EM map of the full virus allowed atomic modeling of residues 30–571 for subunit A, and 20–570 for subunits B and C. The VP1 overall structure consists of an NTA (residues 1–65, Fig 3A and 3B, purple) located inside the capsid, an S domain (residues 66–229, Fig 3A and 3B, blue) that forms the stable inner capsid shell, and a protruding domain (P, residues 238–579) that form the flexible spikes on the surface. The S domain is folded as a

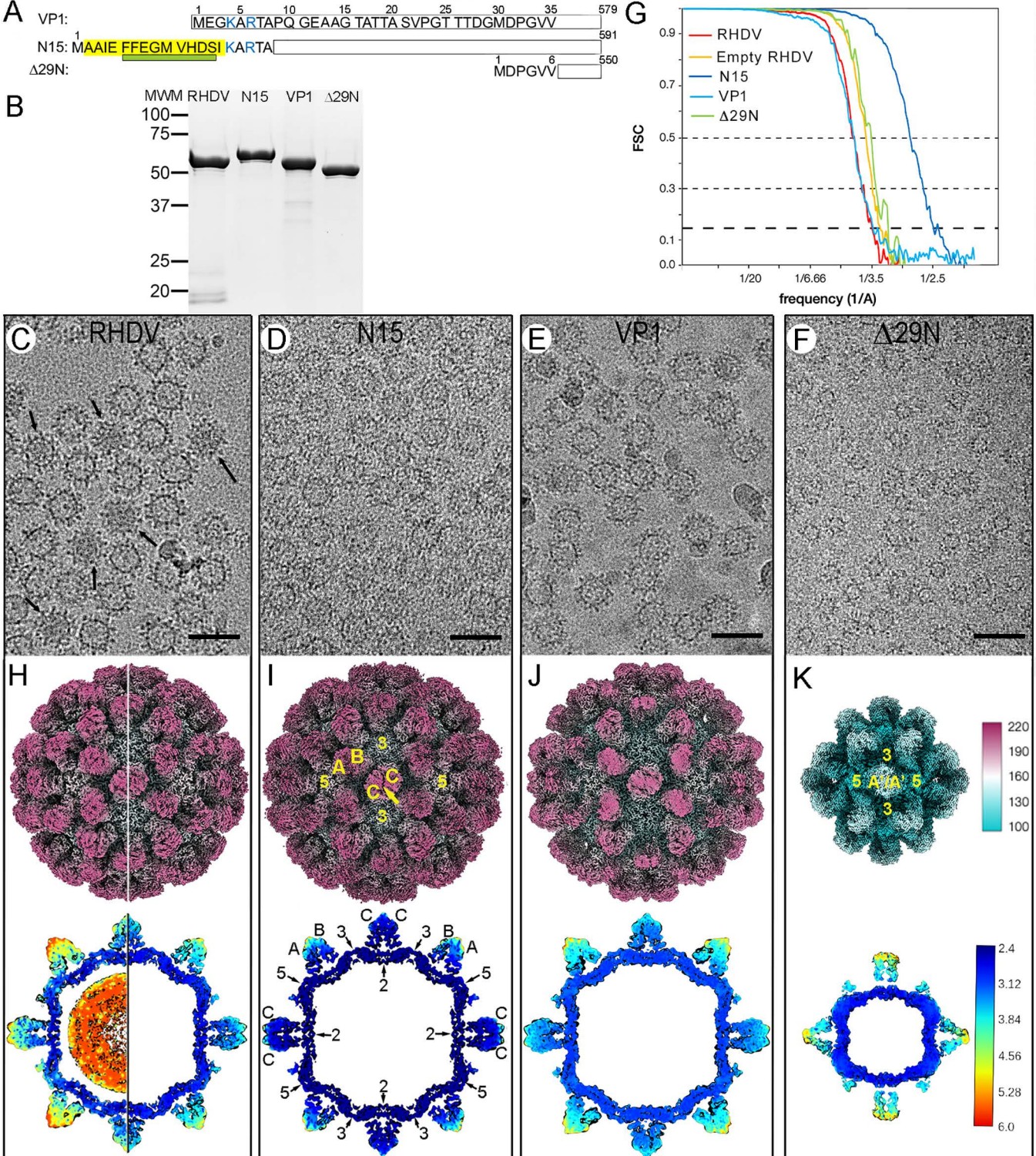

**Fig 1. Biochemical and cryo-EM analysis of RHDV virions and VP1-related VLPs. (A)** Scheme showing wt VP1, VP1 insertion mutant N15, harboring a T-cell epitope of the FMDV nonstructural protein 3A (yellow; the α1 helix is indicated as a green rectangle), and the VP1 deletion mutant Δ29N, indicating their lengths (right). The first 35 N-terminal residues of VP1 are shown (basic residues, blue). **(B)** Coomassie blue-stained SDS-PAGE gels of RHDV, N15, wt VP1, and Δ29N capsids used for cryo-EM data acquisition. Molecular size markers (MWM; kDa) are at left. **(C-F)** Cryo-EM of

RHDV virions (C), N15 (D), VP1 (E), and Δ29N (F) capsids. Arrows (C) indicate full capsids. Bars, 50 nm. **(G)** Fourier shell correlation (FSC) resolution curves for RHDV and VP1-related capsids. Resolutions based on the 0.5, 0.3, and 0.143 criteria are indicated. For the 0.143 threshold, values for the full and empty RHDV were at 3.3 and 3.2 Å (red and yellow), and those for N15 (dark blue), VP1 (light blue) and Δ29N (green) were at 2.5, 3.3 and 2.9 Å, respectively. **(H)** Isosurface representation of full (left half) and empty (right half) RHDV virions, **(I)** N15 capsid, **(J)** VP1 capsid, and **(K)** Δ29N capsid. A/B and C/C dimers (I) and A′/A′ dimers (K) are indicated, and icosahedral symmetry axis are numbered (I, K). Upper row, the radially color-coded outer surfaces viewed along a twofold axis of symmetry contoured at 2σ above the mean density; the color key (right) indicates the radial distance (in Å) from the particle center. Lower row, local resolution assessment shown on 8 Å-thick slabs contoured at 1.2σ above the mean density to highlight the RNA density. The bar indicates resolution in Å (right).

single eight-stranded β-barrel (the jelly roll β-barrel) with two four-stranded β-sheets facing each other (BIDG to CHEF) tangential to the capsid surface (indicated in Fig 3A). The P domain is further divided into two subdomains, P1 (residues 238–286 and 448–579, Fig 3A and 3B, yellow) and P2 (residues 287–447, Fig 3A and 3B, pink). These domain boundaries match very well with previous analysis with RHDV GI.1 (HYD strain), which was resolved from residue 45 [14], and with RHDV GI.2 (SC2020 strain) [15].

P1 is connected to the S domain by a short hinge (residues 230–237, Fig 3A and 3B, green), and P2 can be considered a large insertion in P1. The P1 subdomain folds into a structure similar to other calicivirus P1 domain structures. The P2 subdomain contains a β-barrel of six β-strands (β13–β16 and β18–β19), a fold also conserved in all caliciviruses. This β-barrel is connected by 7 loops of various lengths involved in antigenicity, and act as a receptor-binding site that tolerates insertions.

Whereas the two S domains of a C/C dimer are relatively flat, S domains in an A/B dimer have an inwardly bent conformation relative to the P domain, ~30º and ~13º for A and B S domains, respectively (Fig 3B and 3C and S4 Video). Given the swinging of the protrusions, these values are averaged measurements. A/B dimers are located at local twofold axes, and only the A subunits contribute to the pentamers at the fivefold axes. C/C dimers are located at the icosahedral twofold axes. The hexamers at the threefold axis consist of alternating C and B subunits (Fig 3D). On the virus outer surface, there are 32 cup-shaped depressions (12 at fivefold axes and 20 at the threefold axes).

Although the "up" and "down" conformations for the protruding domain have been reported for RHDV GI.2 [15] and human norovirus [20], we observed no conformational differences of the protruding domain between the mature virion and the VLP after alignment of the A-B dimers (S3 Fig). In RHDV GI.2, the protruding domain of the A-B dimers adopts an elevated conformation in virions, in contrast to a conformation in which the protruding domain rests on the shell domain in the VLP.

### Structure of the inserted T epitope in the N15 capsid

The chimeric N15 protein, which contains a FMDV T cell epitope (residues 21–35 of the nonstructural protein 3A) covalently bound at the VP1 N terminus, assembles into a T = 3 particle similar to the empty VP1 capsid, as their radial density profiles were nearly superimposable (Fig 4A). The major difference was noted at the N15 protein shell on the inner surface (radius ~116 Å) (Fig 4A, arrow). To locate this density difference, we calculated a difference map by subtracting N15 from the VP1 capsid. The resulting difference map showed structural features that could be attributed exclusively to the N15 capsid, denoted in yellow on the inner N15 capsid surface (Fig 4B). This extra density, whose morphology is compatible with an α helix, was located on the triangular structures formed by the α1 helix of A, B, and C subunits, *i.e.*, the center of the icosahedral asymmetric unit.

We built a *de novo* atomic model for this cryo-EM difference density, considering that (i) a previous homology model of the FMDV 3A protein N-terminal region suggested an amphipathic α helix for the sequence [25]FFEGMVHDS[33] [43], as well as (ii) bulky side chains corresponding to Phe25, Phe26, and Val30, equivalent to those denoted as Phe6, Phe7 and Val11 in N15. These three residues are involved in hydrophobic interactions; the opposite helix surface is exposed,

PLOS Pathogens

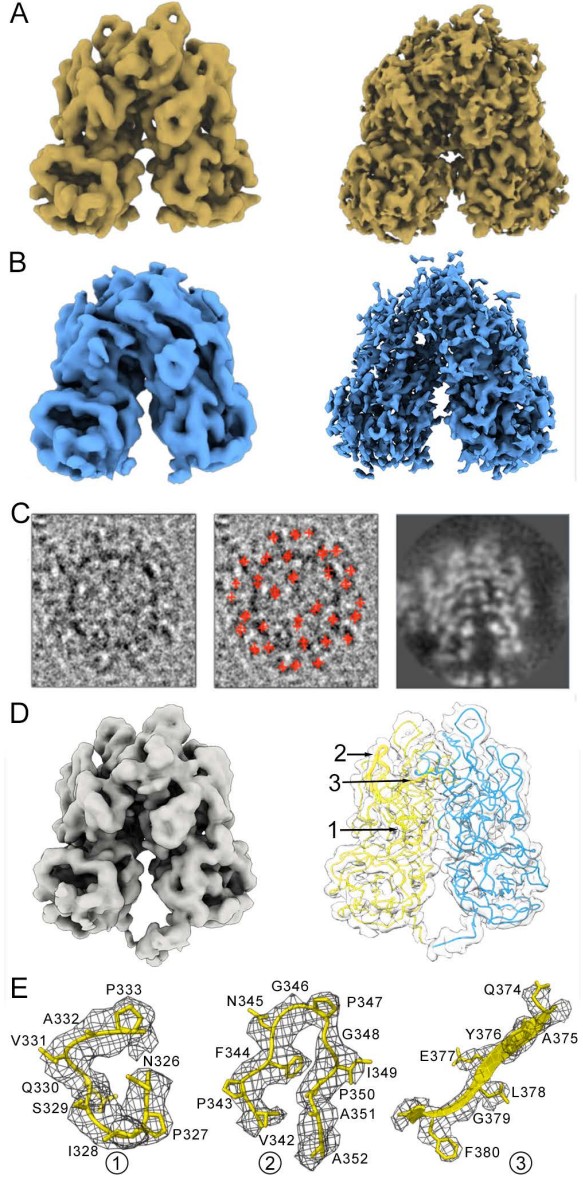

**Fig 2. Near-atomic model of the projecting domain of A and B subunits in the N15 capsid. (A)** Cryo-EM density map of the C/C dimer extracted from the unsharpened (left) and sharpened (right) N15 icosahedral map. **(B)** Cryo-EM density map of the A/B dimer extracted from the unsharpened (left) and sharpened (right) N15 icosahedral map. **(C)** A typical N15 capsid image (left), the same image with marked A/B dimers (red crosses, center), and the central section of the highest resolution map for the A/B dimer after a 3D classification round without angular assignment (right). A/B spikes were extracted in a 97 Å diameter sphere without mask. **(D)** Cryo-EM density map of the extracted, unsharpened A/B dimer (left). The extracted, sharpened map (transparent surface) shows the Cα chain of A (yellow) and B (cyan) P domains (678 residues), highlighting three regions (1-3) within the upper P2 subdomain. **(E)** Regions of the cryo-EM density map of the A P2 subdomain (grey mesh) are shown, with the atomic model of loops 326-333 (left) and 342-352 (center), and the β-strand 374-380 (right).

with mostly acidic residues. Comparison of the 3A α-helix with the VP1 α1-helix showed that the former was thinner than the latter, indicating low 3A α-helix occupancy. We only located one of the three 3A α-helices that could not be ascribed unambiguously to any of the A, B or C subunits of VP1 (the other two 3A α-helices were invisible or disorganized) (Fig 4C).

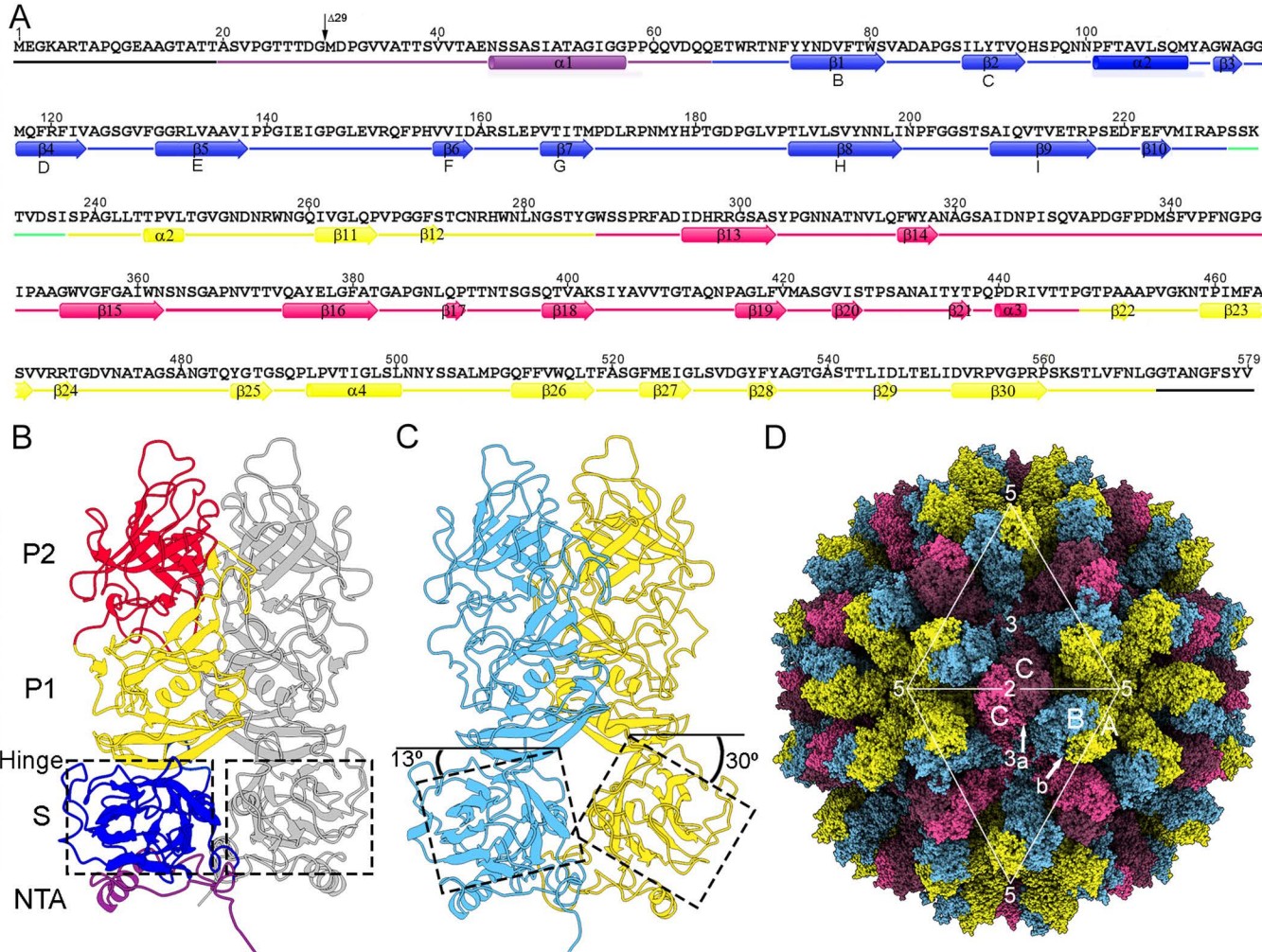

**Fig 3. Cryo-EM near-atomic model of RHDV. (A)** Sequence, secondary structure elements, and domain organization of the 579-residueVP1 (AST89 strain). In the C subunit, the first and last residues assigned were Ala20 and Gly570, respectively. The α-helices and β-strands are represented as cylinders and arrows, respectively, colored distinctly for the four domains: NTA (purple), S (blue), P1 (yellow), and P2 (pink); the hinge between S and P1 is marked (segment 230-237, green), and disordered N- and C-terminal segments are in black. **(B)** Ribbon diagram of the C/C dimer, viewed along the "a" arrow shown in (D). C/C contact between S domains (dashed line) is planar. The C subunit (left) is depicted using the same color scheme as in (A); the other C subunit is in gray. **(C)** Ribbon diagram of the A/B dimer, viewed along the "b" arrow shown in (D). A/B contact between S domains (dashed line) is bent. The A subunit is in yellow and the B subunit in cyan. **(D)** Cryo-EM near-atomic model of RHDV T = 3 capsid. The two VP1 dimer types are indicated (A/B, yellow/cyan; C/C, pink/pink); white triangles define two icosahedral faces. The locations of five-, three-, and twofold axes are indicated.

## Protein-protein and NTA-mediated interactions that stabilize the RHDV virion

We next analyzed the interactions that stabilize the RHDV capsid. Bent A/B and flat C/C dimers are stabilized by intra- and interdimeric interactions mediated by the S and P domains, as well as by the NTA. The VP1 S domains are composed of two structurally differentiated regions, (i) the β-barrel itself with short loops that face the five- and threefold icosahedral axes, and (ii) the large loops at the opposite side of the β-barrel, which include the NTA α1 and S α2 helices, and the segment with the hinge between the P and S domains (Fig 5A, dashed ovals). The NTA α1 helix faces the local threefold

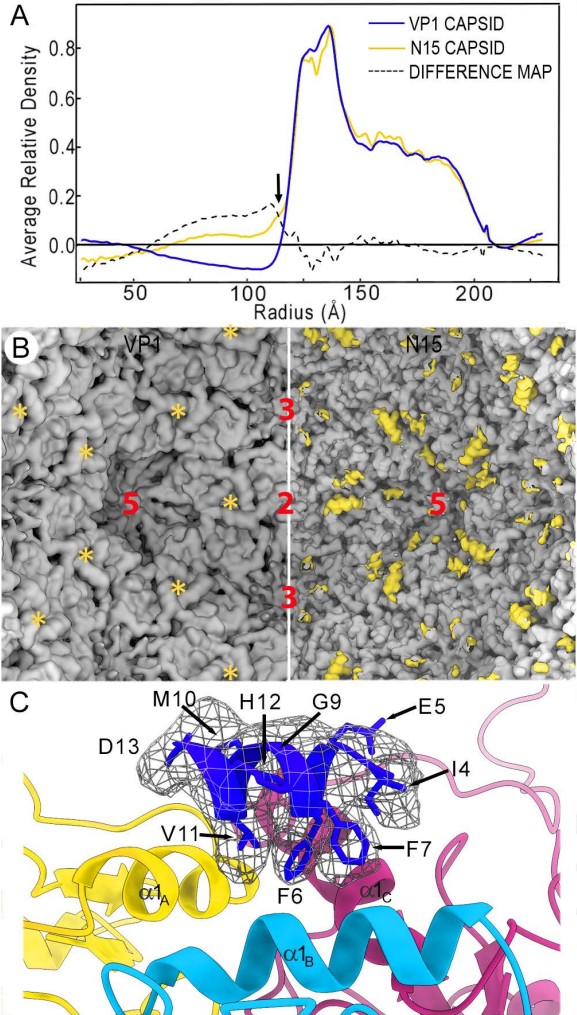

**Fig 4. Structural comparison of VP1 and N15 capsids. (A)** Radial density profiles from the cryo-EM 3D maps of VP1 (blue line) and N15 (yellow line) T = 3 capsids. Protein shells (R = 216-117 Å) are almost superimposable, except for minor differences at the shell interior surface (arrow). A difference map (N15 subtracted from VP1 capsid, dotted line) was calculated by arithmetic subtraction of the density values. **(B)** Difference map calculated by subtracting N15 (right) from VP1 (left) capsid. The resulting difference map, shown as yellow densities, is shown on the inner surface of a N15 capsid; the corresponding empty region on the VP1 capsid is marked (asterisks). **(C)** Difference α-helix (blue) located on top of the three α- helices (contributed by A, B and C subunits of the asymmetric unit). The difference α-helix is included in the inserted T epitope in VP1, segment Ala2-Ile15.

axis in the center of the asymmetric unit, in which three A, B, and C α1 helices interact. The α2 helix faces the local and icosahedral twofold axis to interact with its opposite dimeric partners (Fig 5A).

The molecular switch responsible for VP1 structural polymorphism is located within the NTA in the first 29 N-terminal residues [24]. Although A/B and C/C dimers are the building blocks of the RHDV T = 3 capsid, VP1 pentamers and hexamers are suitable for describing the NTA-mediated interactions beneath the S domains that increase capsid stability (Figs 5B and S4). The structures of the NTA of A, B and C subunits are similar from residue 46 to residue 65 (which includes the α1-helix). The polypeptide chain from residues 30–45 in the A subunit deviates notably compared with that in the B and C subunits, in which the N-terminal end is residue 20 (Fig 5A, left). Most of the ordered NTAs of A subunits traverse beneath

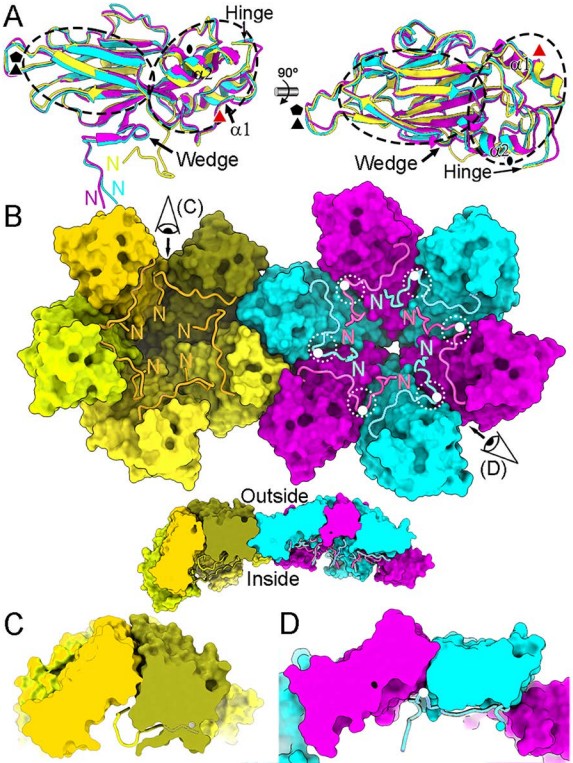

**Fig 5. Structural conformations of NTA regions in the RHDV T=3 capsid. (A)** Superimposed S domains and their NTAs of A (yellow), B (cyan) and C (pink) subunits. The N termini, the α1 and α2 helices of the S domain, the wedge, and the hinge are indicated. Icosahedral and local symmetry axes are indicated (black and red symbols, respectively). **(B)** View from inside the capsid of a pentamer adjacent to a hexamer, both lacking their P domains (surface representations). The segment Met30-Asn45 of NTA from subunits A (yellow colors) around a fivefold symmetry axis is highlighted in orange. Segments Ala20-Thr42 of NTA from subunits B (cyan) and C (pink) are highlighted in bright cyan and pink, respectively, and the Gly29 is shown as a white sphere. The N termini are indicated. The wedge of segment Asp28-Asp31 of B and C subunits is highlighted (white dashed circles). A side cut view is shown of the pentamer adjacent to a hexamer, both lacking their P domains (bottom inset). **(C, D)** Closeup of inset in panel B, showing the interaction surface between two pentameric A subunits (C) and between B and C subunits (D). Note that only N termini of B and C subunits are directed towards the capsid interior. These transversal sections are viewed along the view directions shown in (B).

their cognate S domains and interact with both neighbor A subunits, such that the five A subunits are interconnected at the pentamers. The N-terminal end, segment Met30-Gly33, is swapped between related VP1 subunits at the fivefold axis. The NTA of B and C subunits similarly increase the interlocking between hexameric B and C subunits, and the molecular swap is formed by the segment Ala20-Thr27 at the N-terminal end, although the B and C ends have slightly different conformations.

This atomic model allowed us to visualize how the molecular switch functions. Whereas the first ordered residue is Met30 in the A subunits, the segment Asp28-Asp31 has an ordered structure around the icosahedral threefold axis for B and C subunits, and acts as a wedge that precludes the bending of the protein surfaces in contact, resulting in a fairly planar contact (Fig 5A, 5B and 5D).

The extensive interactions mediated by the NTA of the A, B, and C subunits were visualized in detail in the virion structure, in accordance with previous studies [14,15]. In addition to interactions between the NTA with its cognate S domain in A subunits, the NTA segment Val41-Glu44 interacts with the segment Gly33-Val35 of other fivefold-related A subunit's NTA, establishing a star-like structure around the fivefold axes (Fig 6A and S5 Video). In the B subunit's NTA, the segment Val34-Ala36 binds to the G β-strand of its S domain via a β-augmentation mechanism and, mediated by segment

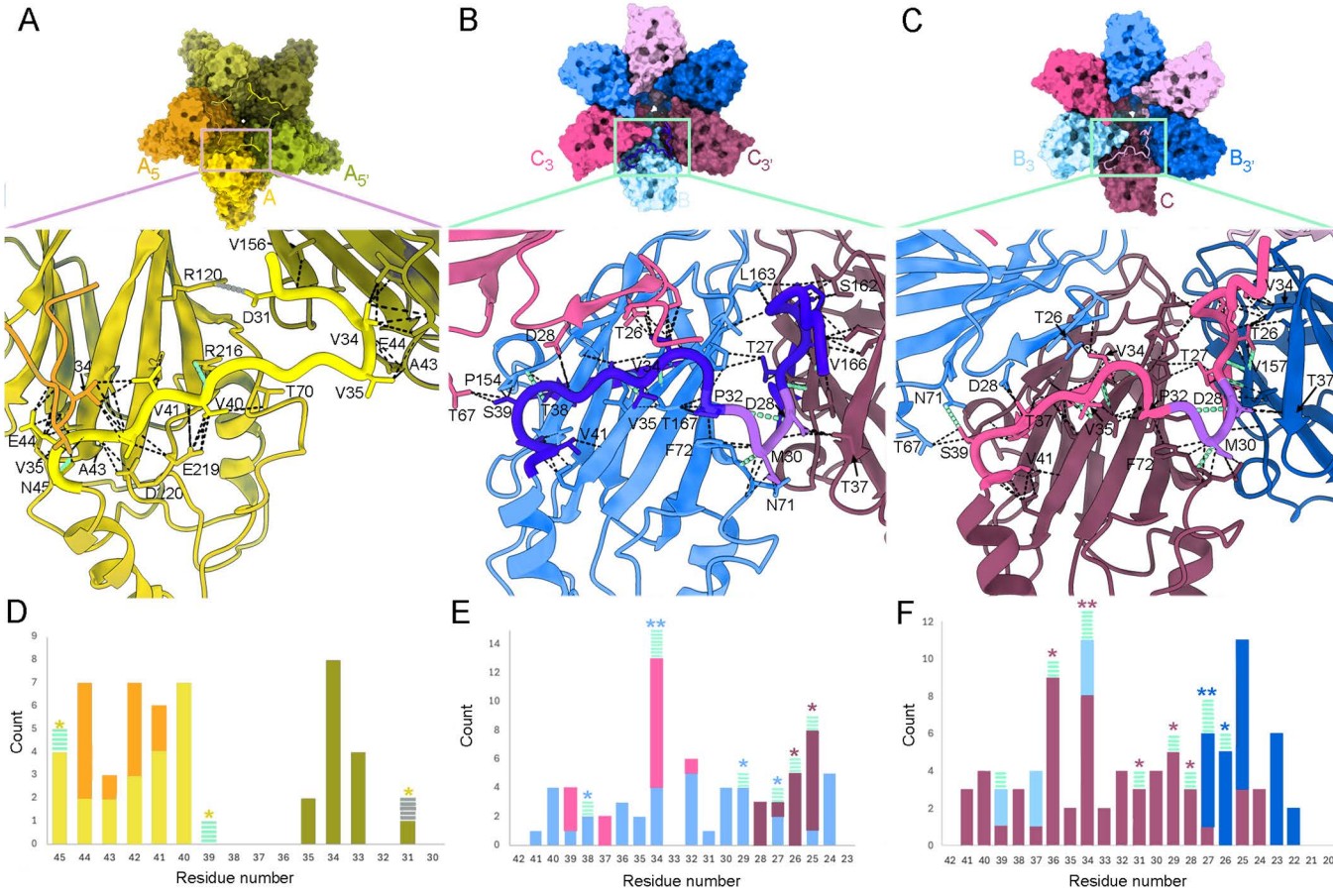

**Fig 6. The NTA is an intrinsic molecular switch and mediates molecular swapping in VP1 pentamers and hexamers. (A)** A VP1 pentamer showing one NTA, segment Met30-Asn45 (yellow). Subunits with subscripts are related to A by a fivefold icosahedral symmetry axis. The inset (bottom panel) corresponding to the boxed region shows a closeup view of the interactions of segment Met30-Asn45 with its cognate A subunit and with $A_5$ and $A_{5'}$ subunits. Van der Waals/hydrophobic interactions, hydrogen bonds and a salt bridge are indicated (dashed black, green lines, grey lines, respectively). **(B, C)** A VP1 hexamer showing one NTA segment of the B (B) or C subunit (C) that includes sequences Ala20-Thr42 (dark blue and pink, respectively); segment Asp28-Asp31 of B and C subunits is highlighted (purple). Closeup views of their interactions are shown in the insets (bottom panels) following a description similar to (A). **(D)** Histogram showing the van der Waals/hydrophobic interactions (solid bars), hydrogen bonds (green dashed bars) and a salt bridge (grey dashed bar) formed by the A subunit NTA segment with its cognate subunit A and with the adjacent $A_5$ and $A_{5'}$ subunits (defined in panel A). Bar colors indicate the subunit that interacts with the A subunit NTA (orange for subunit $A_5$, yellow for its cognate subunit A, and green for subunit $A_{5'}$). Asterisk color indicates the subunit with which hydrogen bonds and salt bridges are established. **(E)** Histogram showing the interactions formed by the B subunit NTA segment with its cognate subunit (bright blue) and with the adjacent subunits $C_3$ (pink) and $C_{3'}$ (purple), as for panel D. (F) Histogram showing the interactions formed by the C subunit NTA segment with its cognate subunit (purple) and with the adjacent subunits $B_3$ (bright blue) and $B_{3'}$ (dark blue), as for panel D.

Thr25-Asp28, interacts with the F β-strand (156–159) of an adjacent C subunit (Fig 6B and S6 Video). There are also numerous contacts with the adjacent C subunit's NTAs on both sides (Fig 6B). In the NTA of C subunits, the residues Ser39, Thr37 and Val34 interact with Thr67 and Asn71, Asp28, and Thr26 and Pro23 of an adjacent B subunit, respectively; the residues Val34 and Ala36 align with its G β-strand, Thr26-Thr27 interacts by β-augmentation with the F β-strand of the other adjacent B subunit, and C Asp28 with B Thr37 and Thr26, Pro23 and Val22 of C subunit's NTA with Val34 of B subunit's NTA (Fig 6C and S7 Video). NTA-NTA interactions are thus implicated in the interactions among A, B and C subunits, and the jelly roll β-barrel of B and C subunits is a nine-stranded β-barrel.

## RNA-protein interactions in the RHDV virions

Radial density profiles of the 3DRs for full and empty RHDV particles were nearly superimposable at the protein shell (r = 113–202 Å) (Fig 7A). Full particles contain an additional density inside the capsid (r ≤ 113 Å) that corresponds to the packaged RNA genome (Fig 7A, orange line). The RNA density is lower than that of the maximum of the protein shell; although the innermost RNA shell appears diffuse, two outer layers are discernible (Fig 7A, arrows), as reported for other ssRNA viruses [44–46]. In the empty capsid, the internal density matched that of the surrounding solvent (Fig 7A, green line). The calculated difference map (full subtracted from empty capsid) highlighted the genome densities (Figs 7A, dashed line and S5).

Analysis of the unsharpened map of the RHDV full capsid showed that, around the threefold axis, the inner surface of the protein shell contacted the underlying density shell, which corresponds to the encapsidated viral genome (S6 and S7 Figs). These connections, 120 per capsid, with a stalactite-like appearance, are formed by the ordered N-terminal ends of the B and C subunit NTA, which have nearly identical conformations, in which the first ordered residue of B and C subunits is Ala20. The six NTAs of B and C subunits surround a poorly defined, jellyfish-like density located at the threefold axis (S7 Fig, orange), similar to that previously described in the RHDV G.II system [15].

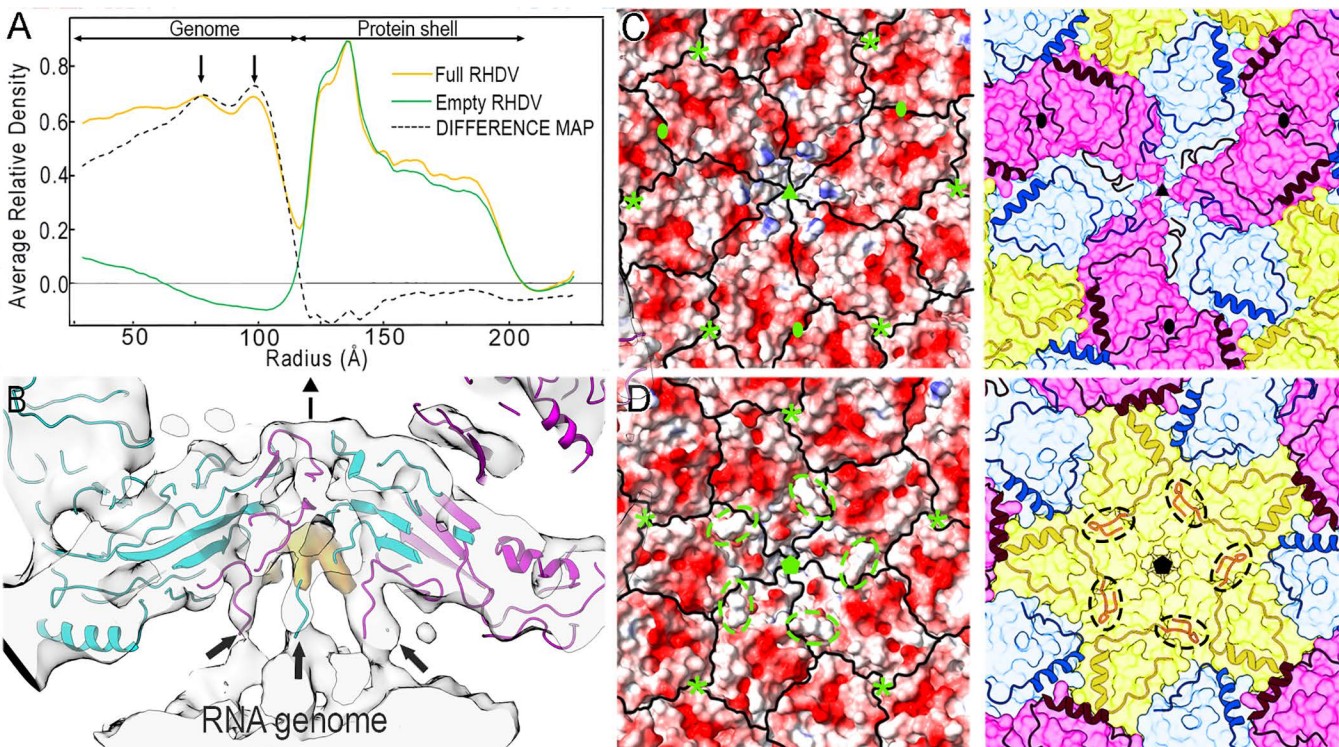

**Fig 7. NTA-mediated interactions with viral genome inside the capsid. (A)** Radial density profiles from the cryo-EM 3D maps of full (orange line) and empty (green line) RHDV capsids. Protein shells (r = 113-202 Å) are almost superimposable. The difference map (full subtracted from empty capsid) shows the genome densities (dashed line); whereas the innermost RNA density is diffuse, two outer concentric layers are discernible at radii of ~77 and ~98 Å (arrows). **(B)** A ~30-Å-thick section of the asymmetric map obtained by symmetry relaxation of the hexameric regions in from the full RHDV particles (the hexamer center is indicated by a black triangle). In this view, three of the six capsid-viral genome contacts mediated by N termini of B and C subunits NTA are indicated (arrows). B, and C subunits are shown in cyan and pink, respectively. The map is contoured at 1σ above the mean density (see S8 Fig for details). **(C, D)** RHDV capsid inner surface viewed down a three- (C) and fivefold (D) axis, represented with electrostatic potentials, showing the distribution of negative (red) and positive (blue) charges, and boundaries for subunits outlined in black (left). Equivalent views showing subunit boundaries and the NTA regions are shown on the right. Note the positively charged stalactites defined by the NTA segments around the threefold icosahedral axis (C, left). Dashed ovals highlight the location of segment 30-38 of A N termini. Symbols indicate icosahedral symmetry axes (asterisks indicate local 3-fold axes in which three α$_1$-helices interact).

We reanalyzed the full RHDV data using 3D classification with symmetry relaxation and systematic focused refinements of the hexameric regions, extracted as independent subparticles, to resolve structural heterogeneity at the threefold axes of symmetry. This analysis yielded cryo-EM maps with variations in the number of NTA-mediated connections, indicating that they are flexible and/or transient (S8 Fig). This approach allowed us to isolate hexameric arrangements with six well-resolved connections that bridge the outer surface of the outermost layer of the genome with the inner surface of the protein shell, which made up 62% of the hexamers included in this analysis (Fig 7B).

The interaction with the viral genome is responsible for the higher ordering of the B and C N-terminal ends compared to that of A subunits. Based on this finding, we propose that in addition to the inherent segment 28–30 of VP1, the viral genome itself acts as a molecular co-switch responsible for the VP1 structural polymorphism.

The electrostatic potential on the capsid inner surface showed an overall negative charge around the five- and threefold axes, which maintains separation from the nucleic acid density (Fig 7C and 7D). This separation is maintained, except at the direct connections between the RNA and the capsid associated with the ordered conformation of the VP1 N termini. Although the viral genome does not interact with the NTA of A subunits, its presence in the full virions influences ordering of the A N terminus, as the subunit A segment 30–38 is disordered in the empty virions (Fig 7D, right), as well as in the VP1 and N15 capsids.

## Mechanical properties of RHDV virions and empty virus-like particles

Using atomic force microscopy (AFM), we analyzed whether differences in viral genome content, *i.e.*, empty versus full capsids, influence viral properties such as capsid stiffness in physiological conditions. RHDV virions were first purified by ultracentrifugation in a sucrose density gradient. Five fractions were collected from top to bottom, and virions were recovered mainly in the middle fractions (fractions 2–4, termed F2, F3, and F4), which were analyzed individually.

RHDV particles showed a similar surface topography (Fig 8A–C, insets) and capsid height [$37.2 \pm 1.4$, $37.4 \pm 1.4$ and $37.5 \pm 1.4$ nm (mean $\pm$SD) for virions of F2, F3 and F4, respectively], indicating minimal capsid deformation after surface adsorption. Fig 8A–C shows a gallery of rigidity slopes obtained from indentation experiments with individual intact RHDV capsids. The histograms summarize the slope distribution of 226 indentations obtained for 38 F2 particles, 222 indentations for 37 F3 particles and 152 indentations for 33 F4 particles. Gaussian fitting of these data yielded a distribution with two rigidity peaks (or spring constants, k), which correspond tentatively to the signature of empty and full virions, with a k value for the empty capsid lower than that of the full capsid. Spring constants were similar for F2 ($0.21 \pm 0.04$ and $0.42 \pm 0.07$ N/m), F3 ($0.20 \pm 0.03$ and $0.42 \pm 0.04$ N/m) and F4 ($0.22 \pm 0.03$ and $0.43 \pm 0.06$ N/m). The ratio between the numbers of events for each rigidity peak (high versus low) was variable; 29% for F2, 16% for F3, and 66% for F4, respectively. These ratios were in accordance with our observations by cryo-EM, in which 18% of the total particles are full (Fig 1C and S1 Table).

RHDV particles were also purified by ultracentrifugation in a CsCl gradient, which yielded a homogeneous population of full virions. Equivalent measurements and calculations rendered 152 indentations for 26 particles (Fig 8D). For empty T = 3 particles from the expression of wt VP1, we obtained 125 indentations for 21 particles (Fig 8E). Although full and empty capsids shared a similar morphology (Fig 8D and 8E, insets) and capsid height ($37.5 \pm 1.5$ and $36.9 \pm 1.6$ nm for full virions and empty T = 3 capsid, respectively), the rigidity of full virions was significantly higher than that of empty T = 3 capsids with k values of $0.43 \pm 0.10$ and $0.22 \pm 0.05$ N/m, respectively. These results confirmed our initial assumption based on the analysis of RHDV purified by sucrose density gradient ultracentrifugation, and highlighted the mechanical role of packed genomic ssRNA in RHDV virions, as empty T = 3 capsids are softer than full T = 3 capsids.

We also analyzed the empty T = 1 capsids, built with almost the same building block as the empty T = 3 capsid. The histogram from T = 1 capsids, with a capsid height of $22.3 \pm 1.6$ nm, summarizes the slope distribution of 88 indentations obtained for 15 particles (Fig 8F). Gaussian fitting of these data yielded a spring constant of $0.17 \pm 0.04$ N/m, similar to empty T = 3 capsids.

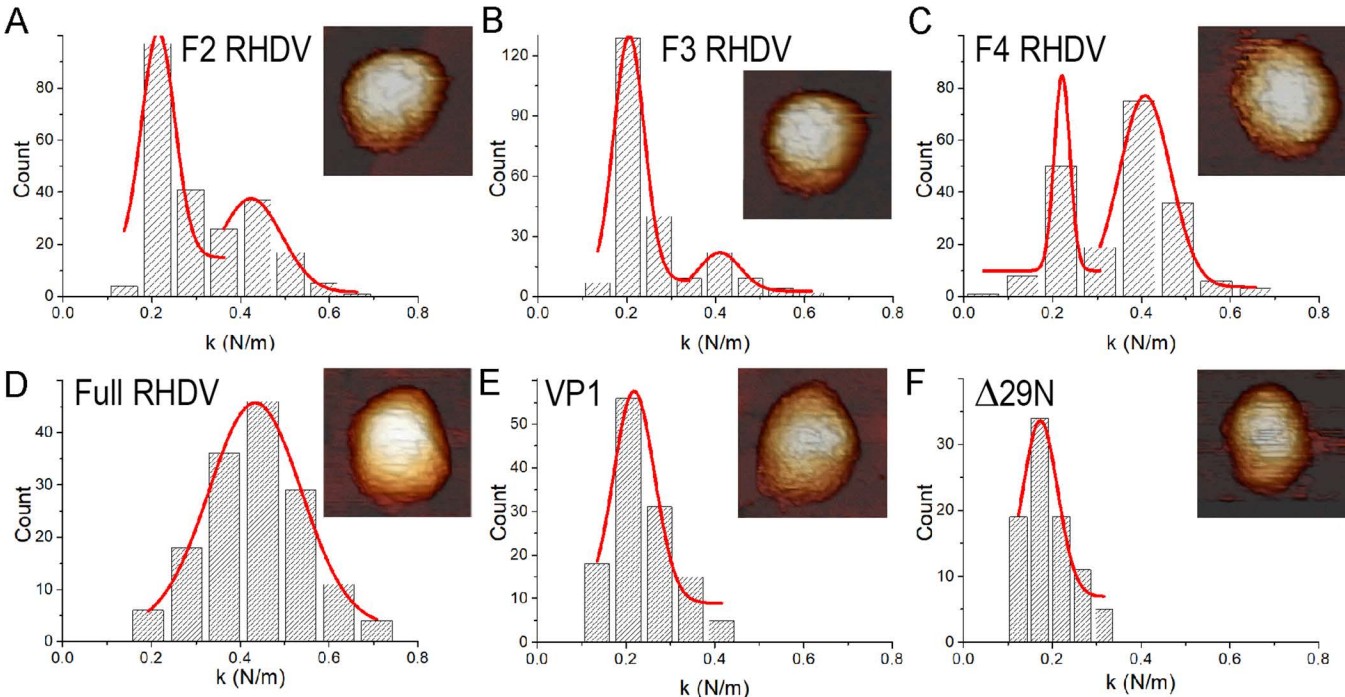

**Fig 8. Mechanical rigidity of VP1 T = 3 and T = 1 capsids. (A-F)** Histogram of slopes of the indentation curves carried out for (A) F2, (B) F3 and (C) F4 RHDV (fractions of full virions obtained after ultracentrifugation in a sucrose gradient), (D) RHDV full virions purified after CsCl gradient ultracentrifugation, (E) VP1 T = 3 empty capsids and (F) Δ29N T = 1 empty capsids. They show the rigidity values (spring constant, k) for individual particles after nanoindentation. AFM images of individual RHDV particles (A-D), empty T = 3 capsids (E) and T = 1 capsids (F) are shown (top, right inset).

## Discussion

In the 180-VP1 subunit T = 3 capsid of caliciviruses, the three different conformations of VP1 (A, B and C) are organized in 30 flat C/C dimers and 60 bent A/B dimers. The three-domain organization of VP1, NTA-S-P, is conserved among caliciviruses. The S domain participates in the icosahedral contacts, the P domain is involved mainly in the dimeric contacts, and the NTA forms a distinct network of interactions beneath the S domains that increases capsid stability. As RHDV cannot be propagated in cell culture, the present study is restricted to structural and biophysical analyses. Our interpretations should thus not be regarded as direct functional evidence, but rather as structural insights that may guide future functional studies.

In noroviruses, the best structurally characterized caliciviruses [16,20], the B subunit NTA is structurally more ordered than those of A and C subunits, and its N-terminal residues interact with the F β-strand in the S domain of the neighboring C subunit around the icosahedral threefold axis (Fig 9A). These interactions could stabilize the flat conformation of the C/C dimer. In vesiviruses [22,23], all three NTA are ordered similarly. The NTA of A, B and C subunits run beneath the S domains of their dimeric partners, forming extensive domain-swapping interactions. Whereas the A and C subunit NTAs converge at the icosahedral threefold axes, the B-NTA extends toward the fivefold axis and interacts with the other fivefold-related B subunit NTAs to form a ring-like structure (Fig 9B). In sapoviruses [21], B and C subunit NTAs interact with adjacent subunits around the icosahedral threefold axis, and no NTA network was identified around the fivefold axis (Fig 9C). In lagoviruses (this study), B and C subunit NTAs show a network of interactions around the threefold axes, similar to that of sapo- and vesiviruses; the A subunit NTA is relatively ordered and is involved in numerous interactions with adjacent subunits around the icosahedral fivefold axis forming a star-like structure (Fig 9D).

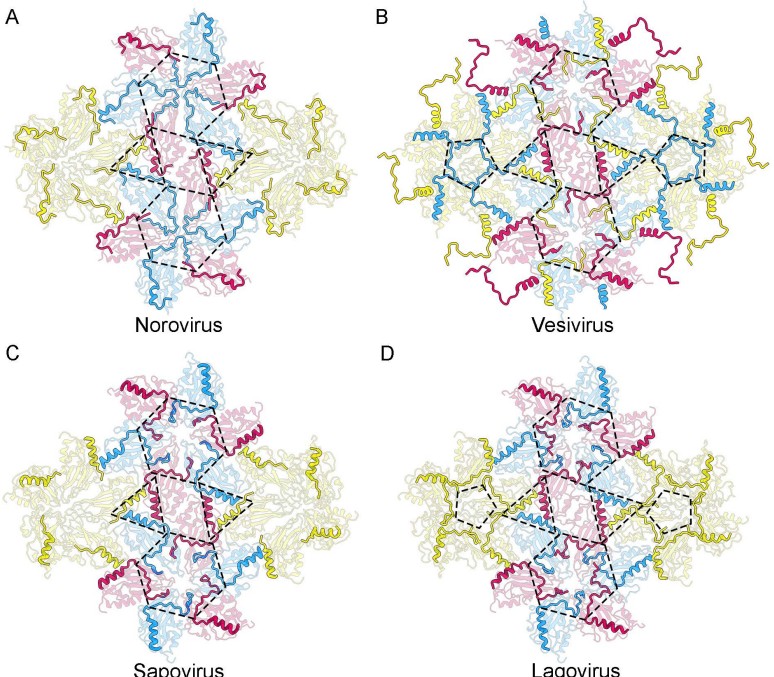

**Fig 9. The NTA network among caliciviruses follows a similar structural pattern. (A-D)** The network of interactions among NTA of subunits A (yellow), B (cyan), and C (pink) in the (A) norovirus, (B) vesivirus, (C) sapovirus, and (D) lagovirus, viewed from inside the capsid. NTA are highlighted and S domains are dimmed in their corresponding colors (P domains are not shown). The dashed triangles delimit the α1-helices from subunits of the same asymmetric unit at the threefold axes. The dashed hexagons and pentagons highlight the network of interactions at the icosahedral three- and fivefold axes, respectively.

The variations described for the NTA ordering/disordering in different caliciviruses could be linked to the presence or absence of viral genome (and/or emulating host RNA molecules). In our RHDV virion structure, the NTA of B and C subunits interact with the viral genome and acquire a structure that explains the molecular mechanism of the VP1structural polymorphism, in which the 28–31 NTA segment is inserted between the adjacent S domains around the threefold axes to make coplanar contacts. This wedge-like structure is conserved and matches similar segments in the vesivirus SMSV (segment 163–167) [22] and human sapovirus (segment 27–31) [21]. The A subunit NTA does not interact with the genome in RHDV, and is more disordered than B and C subunit NTAs, allowing bent contacts between adjacent S domains of A subunits around the fivefold axes. In the SMSV structure, also calculated for native virions by X-ray crystallography analysis, the B subunit NTAs around the fivefold axes form a ring-like structure similar to the star-like structure of RHDV (except it is built by the A subunit NTAs). Direct comparison of full and empty RHDV capsids shows that although there is no direct contact between them, the packaged viral genome induces a certain degree of ordering in the A subunit NTAs, visible from Met30 in the full particles (when the ssRNA is absent, the first ordered A subunit NTA residue is Thr38). In VLPs of noroviruses and sapovirus, the A subunit NTAs are highly disordered at the fivefold axis, as in the empty RHDV VLPs, and there is no network of NTA-mediated interactions [16,21]. Our findings emphasize the importance of the genomic RNA; however, since most human calicivirus structures are derived from empty VLPs, further studies are needed to determine whether RNA fulfills a similar co-switching role.

With the exception of the NTA molecular swapping in vesiviruses, the NTA in different caliciviruses follows a similar organization, in which (i) the three NTA α1-helices from adjacent subunits have hydrophobic interactions at the local threefold axis, and colocalize to the inner center of the asymmetric icosahedral unit, (ii) NTA ordering/disordering is modulated

by the viral genome, and (iii) NTAs establish an extensive network of interactions between S domains at the three- and fivefold axes, which stabilizes hexamers and pentamers, respectively.

Co-assembly of capsid proteins and genomic nucleic acid(s) is extensive among virus families of the picornavirus-like lineage, in which caliciviruses are included (reviewed in reference [47]). In this co-assembly mechanism, the calicivirus ssRNA would nucleate VP1 subunits around itself, a cooperative condensation driven by electrostatic, genome-capsid protein interactions. This genome function is emulated by host nucleic acids that are packaged, as in nodavirus [48] and picobirnavirus [49] VLP assembly. The N-terminal end of VP1 contains only two proximal basic residues, Lys4 and Arg6, which are probably involved in interactions with the viral genome; this attractive interaction could be negatively modulated by two nearby acidic residues, Glu2 and Glu12. Further studies are needed to assess the potential contribution of sequence-specific interactions in addition to electrostatics. The calicivirus NTA contrasts with many NTA of animal and plant viruses with a T=3 capsid, in which basic residues are predominant. As postulated previously [16], the highly basic protein VP2, a 117-residue minor structural protein for RHDV, might have a role in genome condensation (VP2 is also involved in the assembly of a portal-like structure following receptor engagement in FCV [12]).

VP1 dimers are probably the initial building capsomers that nucleate on the viral genome. In the viral factory, and based on their interaction mode with the viral genome, the dimers might coexist in the two conformational states seen in the capsid structure. Whereas a symmetric C/C dimer is formed when both NTA interact with the genome, an asymmetric A/B dimer is formed when only the B subunit does so. This model implies that at the onset of capsid assembly, and in the absence of contacts with the ssRNA, the NTA of VP1-VP1 dimers remain disordered. This suggests a relaxed conformational state resembling the A'/A' dimer described in the Δ29N capsid, which is built from a deletion mutant that lacks the first 29 N-terminal residues. A/B and C/C dimers associate in each other's presence. Capsid assembly might proceed through association of pentamers of A/B dimers that could act as nucleating centers after joining additional C/C dimers; alternatively, the assembly intermediates could be hexamers of three C/C dimers and three A/B dimers. This assembly pathway is conceptually reminiscent of packaging signal-mediated assembly of the T=3 MS2 bacteriophage [50]. Likewise, as observed in HIV assembly [51], interference with intermediate assemblies through VP1-genome interactions (or the NTA network promoted by the genome) might provide a conceptual framework for exploring antiviral strategies against human caliciviruses such as norovirus and sapovirus.

3DFlex and subparticle analyses of T=1 capsids indicate that the protruding spikes undergo a coordinated swinging motion of the VP1 dimers, restricted by the geometry of the hinge regions and occurring perpendicular to their orientation. This collective flexibility likely represents an intrinsic property of calicivirus capsids that balances structural stability with conformational adaptability. In contrast, 3DFlex analysis of both empty and full T=3 capsids revealed similarly small swing angles (~5°), indicating limited protrusion mobility regardless of RNA presence. This reduced motion likely reflects intrinsic structural constraints of the T=3 capsid, although NTA–RNA interactions may contribute additional stabilization by anchoring the inner capsid layer to the packaged genome.

Our AFM studies show that the empty RHDV capsid, whether emptied virions or T=3 VP1 particles, is softer than the full native capsid. The packaged ssRNA is responsible for the stiffening observed for other (mainly dsDNA) viruses such as bacteriophage ϕ29 [52]. Although AFM data suggest that genomic RNA contributes to the stability of the RHDV capsid, we cannot exclude the possibility that VP2, alone or together with the genomic RNA, also participates in this process. The T=3 capsid has a negatively charged inner surface, and the electrostatic repulsion with the packaged genome, together with the RNA-RNA repulsion, would increase stiffness. The direct structural comparison of full and empty virion capsids shows that A subunit NTAs have eight additional disordered residues in empty particles, and lack the network of interactions at the fivefold axes. Our results suggest that pentamers of the full capsid are strengthened by interactions of the A subunit's NTA star-like structure. The stiffening of full virions is probably related to the extracellular step of the viral infectious cycle when viruses are released from the infected cell, but is not a general tendency. Our AFM studies with human picobirnavirus (hPBV) showed that nucleic acid-containing hPBV capsids become flexible when compared with empty capsids [49].

A small empty icosahedron, such as the Δ29N T = 1 capsid, would be anticipated to show greater stiffness than the larger VP1 T = 3 capsid, since both are assembled from nearly identical building blocks. Our results nevertheless indicate that the T = 1 and T = 3 particles show comparable softness; this suggests that the increased disorder at the VP1 N-terminus in the Δ29N capsid (in which the first ordered residue is Gly56) compensates for the difference in capsid size. Further studies are in progress to analyze the mechanical properties of distinct chimeric VP1-based capsids for future applications such as protein containers and/or vaccine vehicles.

Our high resolution cryo-EM structures of the RHDV GI.1 virion and VP1-related VLP provide a notable advance in our understanding of virus assembly of caliciviruses. In addition to the inherent molecular switch of the capsid protein VP1 located in the NTA, which have similar roles in many caliciviruses, our structures highlight the viral genome as a major element in structural polymorphism and virus assembly and stability. While RNA and NTA interactions influence the observed polymorphism, the icosahedral packing of the VP1 molecules might be an even more critical determinant, as higher-order molecular interactions are likely to be responsible for the conformational variation among protein domains.

## Materials and methods

### Virus and cells

RHDV virus (Spanish isolate AST89, GenBank accession no. Z49271) was obtained from infected rabbit liver samples stored at -80ºC, collected in a previous study [32].

The VP1-related VLPs were generated using the baculovirus expression system. Recombinant baculoviruses produced from derivatives of *Autographa californica* nuclear polyhedrosis virus (AcMNPV), were propagated in *Trichoplusia ni* High Five cells (H5), grown in monolayer cultures at 28°C in TNM-FH medium (Sigma-Aldrich), supplemented with 5% fetal calf serum (Gibco, Life Technologies, Thermo Fisher).

### Recombinant VP1-derived constructs

The VP1-derived constructs were generated from previously reported recombinant baculoviruses prepared using the corresponding baculovirus transfer vectors. The VP1 wt construct was generated using plasmid pHAPh306Gsopt [35], containing the VP1 gene from RHDV strain AST89. The insertion construct N15 was obtained using vector pNT15 [37], harboring the sequence coding the T-helper epitope AAIEFFEGMVHDSIK, derived from the 3A protein of FMDV [36], fused to the N-terminal end of VP1 protein. The baculovirus expressing Δ29N was prepared from plasmid pNT29 [24], which harbors a deletion mutant version of the VP1 protein gene lacking the sequence corresponding to the first 29 amino acid residues.

### Virion purification

Liver specimens from RHDV-infected rabbits stored at -80ºC were homogenized with a polytron (PT3100 Homogenizer) and brief sonication in distilled water. After treatment with DNAse I (Roche), samples were adjusted to 2% Sarkosyl (sodium N-lauroylsarcosine, Sigma), 5 mM EDTA in PBS-V (0.2 M sodium phosphate, 0.1 M NaCl, pH 6.0), and incubated 12 h at 4ºC. The liver cell lysates were clarified by centrifugation (1000 x g, 5 min) and the supernatants were layered onto a 10 ml 30% (wt/vol) sucrose cushion in PBS-V and centrifuged using a Beckman SW28 rotor at 27,000 rpm for 4 h. The pellets were resuspended in PBS-V, 5 mM EDTA, and extracted once with Vertrel XF (Fluka, Sigma-Aldrich). Samples were then suspended in a solution of CsCl (0.42 g/ml) in PBS-V and subjected to isopycnic gradient centrifugation at 35,000 rpm for 18 h using a Beckman SW55 rotor. Alternatively, after pelleting through the sucrose cushion, samples were centrifuged on a 25–50% linear sucrose gradient at 38,000 rpm for 1 h using a Beckman SW55 rotor. The gradients generated were fractionated and aliquots of each fraction analyzed by ELISA for RHDV capsid protein. Selected fractions were pooled, diluted in PBS-V, and pelleted by centrifugation at 27,000 rpm for 2 h in a Beckman SW28 rotor. Pellets

were finally resuspended in PBS-V containing protease inhibitors (Complete, Roche) and stored at 4ºC. Protein concentrations of samples were determined using the BCA protein assay kit (Pierce, Thermo Scientific) and were analyzed by SDS-PAGE.

## Expression and purification of VP1-related VLPs

Baculovirus-infected H5 cell monolayers were harvested after incubation for 4 days at 28ºC, washed three times with PBS; the resulting pelleted cells were stored at -80ºC. The recombinant self-assembled VLPs were purified following procedures similar to those described for RHDV virion purification. Briefly, infected cells were homogenized with a polytron, treated with DNAse I, adjusted to 2% Sarkosyl, 5 mM EDTA in PBS-V, and incubated 12 h at 4ºC. Cell lysates were clarified by low-speed centrifugation and the supernatants centrifuged using a Beckman SW28 rotor at 27,000 rpm for 2 h. Pellets were resuspended in PBS-V, 5 mM EDTA, extracted once with Vertrel XF, and centrifuged through a 20% (wt/vol) sucrose cushion in PBS-V at 35,000 rpm for 2.5 h using a Beckman SW55 rotor. Finally, samples were centrifuged on a 25–50% linear sucrose gradient, as indicated above for virion purification. Samples were analyzed by SDS-PAGE [24,35].

## Atomic force microscopy

RHDV virions or VLPs were adsorbed on highly oriented pyrolytic graphite (HOPG, ZYA quality; NT-MDT) by incubation of a 30 µl drop of sample for 30 min and washed with buffer. Surface-attached capsids were imaged with a Nanotec Cervantes AFM (Nanotec, Madrid, Spain) in jumping mode, using 0.1 N/m nominal force constant RC800PSA Olympus silicon nitride cantilevers (Olympus, Tokyo, Japan) with a < 20 nm tip radius (15 nm typical) and 18 kHz nominal resonance frequency. The spring constant for the cantilever was calibrated before each experiment by Sader's method [53]. Capsid position was determined at low resolution (128 pixels per 150 nm) and low maximal force (<100 pN). After individual particles were resolved, indentation measurements were performed. To investigate quantitatively the elastic response of the viral shells in which the nanoindentation process is reversible, we recorded single FZ curves. In these measurements, the applied force was obtained as a function of the displacement of the Z-piezo on which the sample was mounted. The applied force was first calibrated by force-distance (FZ) measurements on the HOPG surface next to the capsid. This was followed by a series of four to six successive FZ curves, each one every 5 s, on the top of the particle at a loading rate of ~50 nm/s and a maximal force of ~1 nN, after which the capsid was reimaged. Only those capsids with no observable damage were included in the analysis. Images were processed using WSxM software [54].

## Cryo-EM data collection

Purified RHDV virions and VLPs (~4 µL) were applied onto R2/2 300 mesh Cu/Rh grids with a continuous carbon layer (Quantifoil Micro Tools, Jena, Germany) that was glow discharged for 15 s at 25 mA (K100X, Emitech), and vitrified using a Vitrobot Mark IV cryofixation unit (Thermo Fisher Scientific). Data were collected on a Talos Arctica or an FEI Titan Krios electron microscope (Thermo Fisher Scientific) operated at 200 or 300 kV, respectively, and images recorded with a Falcon III or K2 detector operating in linear or counting mode, using EPU Automated Data Acquisition Software for single particle analysis (Thermo Fisher Scientific). The total number of recorded movies, nominal magnification, calibrated pixel size at the specimen level, total exposure, exposure per frame, and defocus range for each specimen are described in S1 Table.

## Image processing

All image-processing steps were performed within the Scipion3 software framework [55,56]. Movies were motion-corrected and dose-weighted with Motioncor2 [57]. Aligned, non-dose-weighted micrographs were then used to estimate the contrast transfer function (CTF) with Ctffind4 [58]. All subsequent image processing steps were performed using

Relion 4.0 [59–61] and CryoSPARC [62]. Using automated particle picking with Xmipp, a total of 230,852 (RHDV), 124,519 (N15), 166,047 (VP1) and 389,808 (Δ29N) particles were extracted and normalized. The four data sets used for 2D classification were applied for a 3D classification imposing icosahedral symmetry, using an initial model from preliminary data sets low-pass filtered to 40 Å resolution. The best 25,237 (full RHDV), 141,193 (empty RHDV), 50,616 (N15), 43,950 (VP1) and 302,069 (Δ29N) particles were included in the Cryosparc homogeneous refinement, again imposing icosahedral symmetry, yielding maps with an overall resolution at 3.3 Å (full virions), 3.2 Å (empty virions), 2.5 Å (N15), 3.3 Å (VP1) and 3.0 Å (Δ29N) based on the gold standard (FSC 0.143) criterion. Local resolution was estimated using MonoRes [40].

To solve the structure of the P domain of A/B dimers in the 3D map of the N15 particles, local areas of these regions in the original images were extracted and treated as asymmetric single particles (or subparticles) following the localized reconstruction method [63], using the LocalRec plugin available in the Scipion software framework [64]. The extracted subparticles (~3.5 x 10$^6$) were subjected to two 3D classifications without angular assignment; a final dataset, containing 164,061 particle images, yielded a map with an overall resolution at 2.9 Å. The map was sharpened with Local Deblur [42], which applies a B-factor correction value based on the local resolution estimation from MonoRes.

Flexibility of protruding domains was evaluated with 3DFlex, which provides explicit models of a flexible protein's motion over its conformational landscape [41].

To identify and classify different arrangements of the hexameric regions in full RHDV particles, local areas corresponding to these regions were extracted from the original images and treated as subparticles, and the C3 symmetry was relaxed. Six classes were reconstructed without applying any symmetry and subjected to focused refinements using a truncated conical mask encompassing the region that includes the NTA-mediated connections. The atomic coordinates derived from the icosahedral map were then fitted with Phenix [65].

## Model building and refinement

The backbone of the three polypeptide chains of VP1 in a single asymmetric unit (subunits A, B and C) of the 2.5 Å N15 cryo-EM map was built *de novo* using Coot [66], except for the flexible P domains of A and B conformers. The initial atomic model of A/B P domains was built *de novo* from the cryo-EM density obtained following the localized reconstruction method. A poly-Ala sequence was entered manually for each conformer and the amino acid sequence for each VP1 was adjusted manually. Fit of the atomic model to the density map was improved by iterative cycles of model rebuilding using Coot [66] and Phenix [65]. Real space refinement was performed in Phenix [65,67] with global, local grid search, atomic displacement, nqh flips, and occupancy parameter options. The final VP1 models were obtained by combining both cryo-EM-derived structures that were refined within the icosahedral asymmetric unit. For that, the unambiguous secondary structure elements in the P1 subdomain of the A subunit from the icosahedrally averaged N15 map were selected for reliable docking of the A/B spike atomic structure derived from the localized reconstruction protocol.

The VP1 structure of N15 was used as a template to build a homology model for RHDV, VP1, and Δ29N capsids. The model was then manually adjusted in Coot to optimize the fit to the density, and refined using Coot and Phenix as for the N15 model.

The quality of the model obtained was assessed with Molprobity [68] as implemented in Phenix [69] and with the Worldwide PDB (wwPDB) OneDep System (https://deposit-pdbe.wwpdb.org/deposition). Refinement statistics are listed in S1 Table. Graphics were produced using Chimera X [70]. PDB validation report can be found in S1 File.

## Model analysis

The electrostatic potential for the RHDV capsid was calculated using the Coulombic surface coloring tool available within Chimera X [70]. Protein-protein interactions were analyzed with the Contacts and H-Bonds tools within UCSF Chimera X and PDBsum [71].

Spherically averaged radial density profiles were calculated for both N15/VP1 capsids and full/empty virus maps, and were normalized and scaled to match the fit between both profiles. Difference maps of N15 subtracted from VP1 capsid, and of full capsid subtracted from empty capsid were calculated by arithmetic subtraction of the density values with Xmipp. For surface rendering of the resulting difference map (shown in Fig 4), small islands of density were filtered out, considering only the major differences at radii corresponding to the protein.

## Supporting information

**S1 Fig. 3DR of RHDV GI.1 and VP1-related VLPs.** Central sections from the 3DR (left half) and 8 Å–thick slabs (right half) contoured at 1.2σ above the mean density to highlight the RNA density (as shown in Fig 1H–K, lower row). In central sections, brigther shading indicates higher density. The central transverse sections of the 3DRs highlight internal structural differences, particularly emphasizing the density attributable to the genomic RNA in full RHDV virions.
(TIF)

**S2 Fig. Protrusion spikes swing relative to the shell base.** (A) Analysis of Δ29N dimeric spikes leads to five major classes that include 100% of particles (percentages indicated relative to total selected subparticles). The density maps of the different classes (grey) are superimposed with a spike extracted from the Δ29N capsid calculated after imposing icosahedral symmetry (blue transparent surface). Classes I and V show the spikes with the highest swing angles in opposite directions. (B) Superposition of VP1 dimers fitted in the classes I (grey) and V (red). The swing angle between these two VP1 dimers is ~27°.
(TIF)

**S3 Fig. Structural superimposition of the A/B dimers of the RHDV GI.1 capsid.** Alignment of S domains of A/B dimers in RHDV GI.1 virions (left) and VLPs (center) showed the superposition of their P domains (right). Whereas the P domain of subunit A adopts an intermediate position between those observed in virions and VLPs in RHDV GI.2 D (displaced by ~4 Å relative to each other), the position of the P domain of subunits B is close to that observed in RHDV GI.2 VLP.
(TIF)

**S4 Fig. A pentamer adjacent to a hexamer in the RHDV T = 3 capsid.** The VP1 pentamer and hexamer lack their P domains and are viewed from inside the capsid (ribbon representations), as in Fig 5B. The A subunit NTA (yellow-green colors) around a fivefold symmetry axis is highlighted in orange. The B and C subunit NTAs are highlighted in bright cyan and pink, respectively, and the Gly29 is shown as a black sphere. The N termini are indicated. The wedge of segment Asp28-Asp31 of B and C subunits is highlighted (black dashed circles). Icosahedral symmetry axes (black symbols) and a local twofold axis (red oval) are indicated.
(TIF)

**S5 Fig. Radial density profiles of 3DRs of full and empty virions, and VP1 particle.** (A) Comparison of full and empty RHDV. Protein shells are assentially superimposable. RNA is located at radii <113Å. The density inside the capsid that corresponds to the packaged ssRNA genome is seen in the radial density profiles from both unsharpened (yellow thick lines) and sharpened (yellow thin lines) 3D maps of full particles. The genome in full RHDV appears as (at least) two concentric shells of density at radii of ~77 and ~98 Å (arrows). In the empty RHDV, the internal density was similar to that of the external solvent (green thick and thin lines for unsharpened and sharpened maps, respectively). The calculated difference map (full subtracted from empty capsid) showed the genome densities (dashed line). (B) Central sections from the 3DR of full and empty RHDV, and VP1 capsid viewed along a twofold axis of symmetry. Upper row, unsharpened central sections; lower row, sharpened central sections. (C) Comparison of empty RHDV and VP1 capsid to show that both profiles are superimposable at the protein shell.
(TIF)

**S6 Fig. Location of the NTAs and the jellyfish-like density in the central sections of full RHDV virions.** (A) Central section taken from the full RHDV 3DR, parallel to but displaced ~8 Å from the central section viewed along a twofold axis (protein and genome are white). The atomic models of VP1(A, yellow; B, cyan, C, pink) are superimposed on the protein density (white). (B) Magnified view of the box in A. The first visible residue of VP1 for B (cyan) and C (pink), Ala20, is indicated in three VP1 molecules (arrows). (C) Central section taken from the full RHDV 3DR shown as in A. (D) Magnified view of the box in C. The jellyfish-like density is indicated.
(TIF)

**S7 Fig. Analysis of the jellyfish-like density and the NTAs of B and C subunits at the threefold axis.** (A) ~50-Å-thick section viewed around an icosahedral threefold axis (side view) of the RHDV virion. A, B, and C subunits are shown in yellow, cyan and pink, respectively. Capsid-viral genome contacts mediated by N termini of B and C subunits' NTA are indicated (arrows). The uninterpreted, jellyfish-like cryo-EM density observed at the three-fold axis in the RHDV virion is highlighted in orange. The icosahedral threefold axis is indicated by a black triangle. (B) Inside view of the region shown in (A) indicating the radii of 11 (dashed circle) and 15 Å (solid circle) at which the jellyfish-like arms and NTAs are located, respectively. The last residues of the NTAs of B and C subunits are shown as cyan and pink spheres, respectively.
(TIF)

**S8 Fig. Analysis by symmetry relaxation and systematic focused refinements of hexameric regions extracted from the full RHDV map.** (A) 3D cryo-EM class averages of the region around the threefold axis of the full RHDV capsid, reconstructed without symmetry. Upper row, ~30-Å-thick slabs (side views) of the six asymmetric maps (I-VI) obtained after symmetry relaxation of the hexameric regions. Lower row, inside view of the same maps, ~30-Å-thick slabs, in which RNA density is not shown for clarity. The contact surfaces between the NTAs of the B and C subunits and the RNA genome density are shown in cyan and pink, respectively. These six classes show arrangements with 6 (classes I and II), 5 (III and IV) and 4 (V) NTA-mediated connections; class VI displays an undefined connection (shown in blue). Classes I-VI made up 12, 13, 17, 20, 17 and 21% of the total subparticles, respectively. The six class averages are contoured at 1σ above the mean density. (B) A ~30-Å-thick slab of the asymmetric map calculated by merging and refining classes I-IV obtained by symmetry relaxation of the hexameric regions in the full RHDV capsid (left, side view; right, inside view). The map is contoured at 1σ above the mean density. The colors of the map are as in panel A.
(TIF)

**S1 Video. Video depicting the flexibility of protruding domains of the T = 1 Δ29N capsids.** The flexibility of the 30 protruding dimers in the Δ29N capsid was analyzed with 3DFlex.
(MOV)

**S2 Video. Video depicting the flexibility of protruding domains of the T=3 capsids (full virions).** The flexibility of the 90 protruding dimers in the T=3 capsids was analyzed with 3DFlex.
(MOV)

**S3 Video. Video depicting the flexibility of protruding domains of the T=3 capsids (empty virions).** The flexibility of the 90 protruding dimers in the T=3 capsids was analyzed with 3DFlex.
(MOV)

**S4 Video. Structure of A/B and C/C dimers.** Structure of the A/B (left) and C/C (right) dimers (ribbon diagrams) extracted from the RHDV GI.1 virion, colored as in Fig 3D. The organization in P1, P2, hinge, S and NTA regions of A/B and C/C dimers is indicated. The A/B contact between S domains (dashed line) is bent (the A subunit is in yellow and the B subunit in cyan). The C/C contact between S domains (dashed line) is planar (the C subunit is in pink).
(MOV)

**S5 Video. Interactions of the A subunit NTA with its neighbor A subunits around a fivefold symmetry axis.** The A subunit NTA (segment Met30-Asn45) is highlighted in orange, and interactions with other fivefold-related A subunit NTA and S domains are shown (color scheme as in Fig 6A).
(MOV)

**S6 Video. Interactions of the B subunit NTA with its neighbor C subunits around a threefold symmetry axis.** The B subunit NTA (segment Ala20-Thr42) is highlighted in cyan, and interactions with other threefold-related C subunit NTA and S domains and its cognate B subunit are shown (color scheme as in Fig 6B).
(MOV)

**S7 Video. Interactions of the C subunit NTA with its neighbor B subunits around a threefold symmetry axis.** The C subunit NTA (segment Ala20-Thr42) is highlighted in pink, and interactions with other threefold-related B subunit NTA and S domains and its cognate C subunit are shown (color scheme as in Fig 6C).
(MOV)

**S1 Table. Cryo-EM data collection and refinement statistics.**
(DOCX)

**S1 File. PDB validation report.**
(PDF)

## Acknowledgments

We thank Rocío Arranz and Javier Chichón of the CNB-CSIC Cryo-EM facility (Madrid) for help with cryo-EM data acquisition, Elisa Torres and Carolina Cabezas at CISA-INIA/CSIC lab for their technical assistance, and Catherine Mark for editorial assistance. We acknowledge the UK national electron Bio-Imaging Centre (eBIC) and the Astbury Biostructure Laboratory for access and support of the cryo-EM facilities.

## Author contributions

**Conceptualization:** Juan Bárcena, José R. Caston.

**Data curation:** Guy Novoa, Carlos P. Mata.

**Formal analysis:** Guy Novoa, Carlos P. Mata, Johann Mertens, María Zamora-Ceballos, Juan M Martínez-Romero, Juan Fontana, Juan Bárcena, José R. Caston.

**Funding acquisition:** Juan Bárcena, José R. Caston.

**Investigation:** Guy Novoa, Carlos P. Mata, Johann Mertens, María Zamora-Ceballos, Juan M Martínez-Romero, Juan Fontana, José L. Carrascosa, Juan Bárcena, José R Castón.

**Project administration:** Juan Bárcena, José R. Caston.

**Supervision:** José L. Carrascosa, Juan Bárcena, José R. Caston.

**Validation:** Guy Novoa, Carlos P. Mata, Johann Mertens, Juan M Martínez-Romero, Juan Fontana.

**Visualization:** Guy Novoa, Carlos P. Mata, Johann Mertens, María Zamora-Ceballos, Juan Fontana.

**Writing – original draft:** Juan Bárcena, José R. Caston.

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
