## [Decision Letter · Decision Letter 0]

31 Aug 2025

Calicivirus assembly and stability are mediated by the N-terminal domain of the capsid protein with the involvement of the viral genome

PLOS Pathogens

Dear Dr. Caston,

Thank you for submitting your manuscript to PLOS Pathogens. After careful consideration, we feel that it has merit but does not fully meet PLOS Pathogens's publication criteria as it currently stands. Therefore, we invite you to submit a revised version of the manuscript that addresses the points raised during the review process.

Please submit your revised manuscript within 60 days Oct 30 2025 11:59PM. If you will need more time than this to complete your revisions, please reply to this message or contact the journal office at plospathogens@plos.org. Please include the following items when submitting your revised manuscript:

We look forward to receiving your revised manuscript.

Kind regards,

Suchetana Mukhopadhyay

Academic Editor

PLOS Pathogens

Alexander Gorbalenya

Section Editor

PLOS Pathogens

Sumita Bhaduri-McIntosh

Editor-in-Chief

PLOS Pathogens

orcid.org/0000-0003-2946-9497

Michael Malim

PLOS Pathogens

orcid.org/0000-0002-7699-2064

**Additional Editor Comments:**

Dear Dr. Caston and co-authors,

Thank you for your patience as we gathered reviews for your recent submission to PLoS Pathogens.

In this work, you have used complementary structural approaches to obtain a high-resolution structure of intact calicivirus virions, allowing the visualization of vRNA–NTA interactions for the first time. Complementary AFM studies demonstrated that vRNA contributes to virion stability and rigidity. Together, these findings advance our knowledge of calicivirus structure and provide new insights with important implications for infection and viral biology.

That said, there are concerns that should be addressed. In particular, three reviewers (#2-4) note importance and limitations of the provided RNA characterization. For instance, Reviewer #4 raises a key point regarding the need for stronger validation of the density attributed to vRNA, a major conclusion in this work. Their suggestions for additional approaches to strengthen this conclusion, and discussion about assigning the density to vRNA at the sigma level at which this is observed are worth careful consideration. Please refer to the full set of reviewer comments below for further details.

Tuli Mukhopadhyay

**Journal Requirements:**

1) We noticed that you used the phrase 'data not shown' in the manuscript. We do not allow these references, as the PLOS data access policy requires that all data be either published with the manuscript or made available in a publicly accessible database. Please amend the supplementary material to include the referenced data or remove the references.

2) We notice that your supplementary Tables are included in the manuscript file. Please remove them and upload them with the file type 'Supporting Information'. Please ensure that each Supporting Information file has a legend listed in the manuscript after the references list.

3) Please amend your detailed Financial Disclosure statement. This is published with the article. It must therefore be completed in full sentences and contain the exact wording you wish to be published.

2) If any authors received a salary from any of your funders, please state which authors and which funders..

**Reviewers' Comments:**

Reviewer's Responses to Questions

**Part I - Summary**

Reviewer #1: The authors studied RHDV virions and several virus-like particle structures at near atomic resolutions using cryo-EM. The reconstructions achieve much higher resolutions than previously published studies, thereby allowing detailed investigation of specific amino acid residue interactions within the protein building blocks of the virus. The study revealed that the critical molecular components (the NTA network and the viral RNA genome) determine structural polymorphisms of VP1 dimers in an icosahedral capsid. The molecular interactions are compared with those of other caliciviruses, highlighting shared features and differences. The authors also used AFM to study the mechanical stability of both T=3 empty and full particles, as well as the T=1 empty particles. The experiments suggested that RNA not only facilitates particle assembly but also contributes to its stability under mechanical stress. Overall, this is a well-written paper that provides a comprehensive structural analysis of the virus structure, representing a significant advancement in the structural biology of caliciviruses.

Reviewer #2: This study investigated the cryo-EM structures of the Rabbit hemorrhagic disease virus (RHDV) virion and three recombinant RHVD virus-like particles with differing N-termini. The focus of this study was on the role of the N-terminal arm (NTA) in virus assembly, virus rigidity, and interaction with the viral RNA genome. Comparison of the structures revealed high similarity of T=3 icosahedral capsid in the RHDV virion and the full-length virus-like particles, whereas the delta29 capsid lacking the N-terminal 29 amino acids formed T=1 capsids, indicating the importance of the NTA in virus assembly. Evaluation of the N15 capsid containing a FMDV-derived T-cell epitope enabled visualization of the epitope inside the capsid in some cases. The NTA structures differed in A/B/C subunits and provided a molecular basis for their roles in promoting A/B and C/C dimers, and subsequently pentamers and hexamers. Additionally, the RHVD virion electron density maps showed electron density around the ordered N-terminal ends (Ala20) at the 3-fold axis, which the authors attribute to interacting viral RNA genome. Finally, atomic force microscopy (AFM) studies reveal that full virions are more rigid than empty T=3 capsid, which the authors conclude is a “mechanical role” for genomic RNA in RHVD virions.

Overall, this work is consistent with and forms many similar conclusions on the NTA as the previous study on the structures of RHVD2 virions and virus-like particles (Ref. 15). A difference of this work is that it concludes that the extra electron density near the 3-fold NTEs is viral genomic RNA whereas the previous study concluded that this density is from the remaining disordered NTA amino acids 1-19. Also, a novelty of this study is the AFM work that reveals that the genomic RNA contributes to viral rigidity.

Reviewer #3: The manuscript “Calicivirus assembly and stability are mediated by the N-terminal domain of the capsid protein with the involvement of the viral genome” by Novoa et al reports high-resolution cryo-EM structures (2.5–3.3 Å) of rabbit hemorrhagic disease virus (RHDV) virions along with a set of related virus-like particles (VLPs), combined with atomic force microscopy to probe the mechanical properties of capsids. The key finding is that the N-terminal arm (NTA) of the capsid protein VP1 does not function in isolation, but rather requires interaction with the viral RNA genome to stabilize specific conformations of VP1 dimers. These RNA–NTA contacts act as a “molecular co-switch” that guides T=3 assembly and strengthens capsid rigidity. The study nicely integrates structural and biophysical approaches and provides a new way of thinking about how the genome itself contributes to assembly and stability. Some strengths include: 1) the cryo-EM structures of authentic virions at near-atomic resolution are a significant technical achievement, allowing the authors to visualize NTA–RNA contacts directly; 2) the combination with AFM is clever and powerful, showing that RNA packaging makes virions mechanically stiffer than empty shells and thus adding a functional dimension to the structural observations; 3) the conceptual advance is important: while many calicivirus structures exist, this is the first to demonstrate that the genome actively shapes NTA conformations and assembly, rather than being a passive passenger; and 4) he paper is also placed thoughtfully in the broader calicivirus context, with comparisons to noroviruses, vesiviruses, and sapoviruses. Despite the overall strength of the manuscript, it could be improved by addressing a few issues listed below:

Reviewer #4: Authors describe 2.5-3.5A cryo-EM structure of RHDV mature virion, empty virion particles, recombinant VLP assembled from VP1, and recombinant VLPs assembled from a N-terminal truncation. In addition, the authors have also performed AFM studies to investigate whether capsid stability is affected by the presence of RNA by comparing full and empty virions and VLPs. Overall, from these studies, the authors suggest that interactions between the NTA and genomic RNA plays a guiding role

In conferring proper conformational states of VP1 dimers and the capsid assembly.

Although the structural analysis is well-performed and the manuscript is well-written, including all relevant methodological details, and appropriate illustrations and supplementary material.

However, in my opinion, this study falls short in terms of significance and general interest. It is perhaps more suited for a specialized journal. The cryo-EM structures of the mature G1.2 RHDV virion, VLP assembled from the full-length VP60, and VLP assembled from the NTA-truncated VP60 of RHDV GI.2 (at resolutions of 2.5 Å, 2.5 Å, and 2.4 Å), respectively, have already been published. I appreciate that the structural analysis in this manuscript focuses on G1.2 RHDV, which has not been published before.

Most of the inferences about the network of NTA in RHDV, as well as other members of the Caliciviridae, have been thoroughly examined and visualized either by crystallography or high-resolution cryo-EM. One unique aspect of this manuscript is the supposed visualization of the effect of genomic RNA on the differential conformation of the NTAs and their interaction with the neighboring subunits.

But this aspect is not convincing at all. It is unclear whether the density supposedly attributed to RNA is indeed that of RNA, particularly considering that the density is visualized in an unprocessed map at a 0.6 sigma level. That level borders on the noise; it could be noise emanating from the local breakdown of icosahedral symmetry, or perhaps even due to VP2, which, in fact, has been shown to contribute to the stability of the capsid in norovirus VLP. A stronger validation of this aspect would have garnered more enthusiasm; this perhaps requires performing symmetry relaxation and systematic focused refinements. AFM studies suggest that genomic RNA may play a critical role in capsid assembly and stability; however, this observation alone is insufficient and non-confirmatory, as it is equally possible that VP2 (alone or in combination with genomic RNA) also plays a role.

Minor:

Reference citations in several places could be improved by providing references in chronological order, based on when a particular structural observation was first made and to which caliciviruses it pertains.

**Part II – Major Issues: Key Experiments Required for Acceptance**

Reviewer #1: 3DFlex analysis shows that the protrusion spikes swing relative to the shell base. This is a very interesting observation that also raises many questions. Some of the following questions could be addressed in the discussion section or explored further if computationally feasible.

What is the nature of the motion? Is it directional relative to the base for different dimers (C/C and A/B)? The VP1 dimer contains two short hinge regions (residues 230-237) in the VP1 dimer, which should limit the swing direction by structural configuration. Is the direction of swing perpendicular to the orientation of these hinges in the dimer? Is there also a twist or swelling relative to the local 2-fold axis? Could there be a local shift of the whole dimer? Could the motion observed in 3DFlex represent a combination of all these movements? Furthermore, if particle size varies, would curvature differences also contribute to apparent motion in 3DFlex?

3DFLex analysis shows that the T=3 capsid (empty) exhibits reduced protrusion swing angles comparing to the T=1 Δ29N particle (~5 degrees vs. 10-15 degrees). Is this reduction due to the stabilizing effect of NTA? Would RNA interactions further stabilize the virus structure? Would 3DFlex analysis of a full virion containing RNA reveal an even further reduction in motion?

Given the observed swinging of the protrusions, the calculated angles of the S domain relative to the P domain shown in the Figure 3 B and C may need to be reconsidered. The ~30d and ~13d for A and B relative to the S domain in the RHDV reconstruction map are likely the averaged measurements of these angles.

One computational approach could be to perform 3D classification using a larger box to extract local dimer regions. If the particles are aligned based on one end of the dimer (either the base or the tip), the 3D classification might reveal distinct classes with different swing angles.

Reviewer #2: The conclusion that the extra electron density near the 3-fold NTEs is viral genomic RNA is not clearly supported. It appears possible that the electron density is from amino acids 1-19, as was concluded in Ref. 15. The authors should compare their electron density maps from the RHVD virion and the empty N15 and VP1 capsid structures in Figure 7 to support their conclusion that this density is only in the RHVD virion, supporting it is viral RNA. The authors should also comment as to why they conclude it is viral RNA and not the NTA, and why their data/conclusion differs from Ref. 15.

Reviewer #3: How specific are the RNA–NTA interactions: The data clearly show RNA density contacting the NTAs, could the authors comment as to whether these are sequence-specific interactions or simply electrostatic. A bit more discussion would help here.

How general is this mechanism: The authors compare across caliciviruses, which is very helpful. But most of the human calicivirus structures come from empty VLPs, so it is hard to know whether RNA plays a similar co-switch role there. It would be useful if the authors more explicitly noted these limits, while still arguing for likely conservation.

Functional assays: The paper is entirely structural and biophysical. This is understandable given that RHDV can’t be propagated in cell culture, but it would strengthen the manuscript if the authors directly acknowledged this limitation, so readers don’t over-interpret the structural results as functional proof.

Reviewer #4: As discussed above

**Part III – Minor Issues: Editorial and Data Presentation Modifications**

Reviewer #1: 1) The P domains adopt flexible conformations relative to the S domains, as shown in Movies S1 and S2. This flexibility limits the overall resolution when whole-particle images and icosahedral symmetry are used for reconstruction. Therefore, masks were applied for localized reconstructions to achieve higher resolutions (Figure 1G). Please indicate the size of the mask in a figure or in a supplementary figure.

2) The paper concludes that the NTA network and RNA interactions determines the structural polymorphisms of VP1 dimers in an icosahedral capsid – specifically, the relative angles between the S and P domains in molecule A, B and C. However, it is important to note that an even more significant factor may be the icosahedral packing of the molecules. Higher order molecular interactions likely play a critical role in driving the conformational variation of the protein domains.

3) For readers unfamiliar with RHDV research, the biological significance of N15 VLPs is not immediately clear. Does the FMDV T cell epitope (residues 21-35 of the nonstructural protein 3A) share sequence or biochemical similarity with the NTD of RHDV VP1? Please provide a clarification in the Introduction or Results section regarding N15 (i.e. the inserted T epitope in the N15 capsid). Additionally, would co-expression of this peptide affect infectivity of the virion?

4) Line 425-426 states “the Δ29N capsid T=1 capsid would also be anticipated to be stiffer than the large VP1 T=3 capsid”. However, since the T=1 capsid lacks the NTA, wouldn’t this make the capsid softer, even though its smaller size might render a stiffer particle? In fact, the measured spring constant for the T=1 particle (0.172 ± 0.039 N/m) is smaller than those of the T=3 full and empty particles (0.433 ± 0.103 and 0.218 ± 0.046 N/m), suggesting that the T=1 particles are softer. So please clarify this statement (line 425-426).

5) It will be very helpful to readers if the figures and movies included more labels. For example:

(a) Movie S3: please label the NTA, S and P1/P2 domains.

(b) Figure 1: please label the sample names in panels C-F

(c) Line 248 states “The molecular switch responsible for VP1 structural polymorphism is located within the NTA in the first 29 N-terminal residues. These molecules are superimposed well”. Please label where the residue 29 is in Figure 5B.

(d) Figure 6D-F: please clarify the meaning of the bar colors (yellow orange, green)

Reviewer #2: 1. Line 122: Incomplete sentence: “…information regarding the spatial organization of the ssRNA within these virions and its interactions with the capsid.” Likely should end with “is lacking.”

2. Many figures would benefit from labels for panels, so the reader doesn’t have to go through the legends (i.e. Fig. 1H-K, Fig. 4A (add legend for the 2 colors), and Fig. 8)

3. Fig. 6DEF legend needs to be clearer in describing the figures, and these figures should be referred to in the text.

4. Lines 307-328: In most cases, there should be 1 significant digit in the standard deviations, and the last significant digit in the reported mean should align with the standard deviation (i.e. 0.214 +/- 0.037 should be reported as 0.21 +/- 0.04).

5. Line 320: Should be Fig. 1C, not 1G.

Reviewer #3: The comparisons to HIV and MS2 are interesting but might mislead non-expert readers into thinking these are mechanistic parallels. It would help to clarify that these are only conceptual analogies.

Try to be consistent with terminology: the manuscript flips between “molecular switch” and “co-switch.”

The supplementary movies (S1–S6) could be described in more detail in the text so the reader knows what they’re meant to illustrate.

Reviewer #4: Overall okay

PLOS authors have the option to publish the peer review history of their article (what does this mean? ). If published, this will include your full peer review and any attached files.

**Do you want your identity to be public for this peer review?** For information about this choice, including consent withdrawal, please see our Privacy Policy .

Reviewer #1: No

Reviewer #2: No

Reviewer #3: No

Reviewer #4: No

**Figure resubmission:**

**Reproducibility:**



---

## [Decision Letter · Decision Letter 1]

17 Nov 2025

Dear Dr. Caston,

We are pleased to inform you that your manuscript 'Calicivirus assembly and stability are mediated by the N-terminal domain of the capsid protein with the involvement of the viral genome' has been provisionally accepted for publication in PLOS Pathogens.

Best regards,

Suchetana Mukhopadhyay

Academic Editor

PLOS Pathogens

Alexander Gorbalenya

Section Editor

PLOS Pathogens

Sumita Bhaduri-McIntosh

Editor-in-Chief

PLOS Pathogens

orcid.org/0000-0003-2946-9497

Michael Malim

Editor-in-Chief

PLOS Pathogens

orcid.org/0000-0002-7699-2064

Editor Comments :

Thank you very much for your patience as we gathered the reviewer comments. Congratulations! We are happy to accept your manuscript for publication. The additional analysis requested during the initial review have improved the depth and quality of the article.

Reviewer Comments (if any, and for reference):

Reviewer's Responses to Questions

**Part I - Summary**

Reviewer #1: (No Response)

Reviewer #2: The authors have addressed all of my comments and concerns. Thank you for performing the additional structural analyses, which adds to the impact and depth of this work.

Reviewer #4: I think the authors have diligently considered all the comments/suggestions made and made serious to attempts to respond to them. Heir responses are satisfactory and I recommend publication.

**Part II – Major Issues: Key Experiments Required for Acceptance**

Reviewer #1: (No Response)

Reviewer #2: None

Reviewer #4: None - all addressed

**Part III – Minor Issues: Editorial and Data Presentation Modifications**

Reviewer #1: (No Response)

Reviewer #2: None

Reviewer #4: None - all addressed

PLOS authors have the option to publish the peer review history of their article (what does this mean? ). If published, this will include your full peer review and any attached files.

**Do you want your identity to be public for this peer review?** For information about this choice, including consent withdrawal, please see our Privacy Policy .

Reviewer #1: No

Reviewer #2: No

Reviewer #4: **Yes: ** B.V.V. Prasad

---

## [Editor Report · Acceptance letter]

Dear Dr. Caston,

We are delighted to inform you that your manuscript, "Calicivirus assembly and stability are mediated by the N-terminal domain of the capsid protein with the involvement of the viral genome," has been formally accepted for publication in PLOS Pathogens.

Best regards,

Sumita Bhaduri-McIntosh

Editor-in-Chief

PLOS Pathogens

orcid.org/0000-0003-2946-9497

Michael Malim

Editor-in-Chief

PLOS Pathogens

orcid.org/0000-0002-7699-2064